# Combinatorial effects on gene expression at the *Lbx1*/*Fgf8* locus resolve split-hand/foot malformation type 3

Giulia Cova [1,2,13,17] ✉, Juliane Glaser[1,17], Robert Schöpflin[1,2,3], Cesar Augusto Prada-Medina[1,14], Salaheddine Ali [1,2], Martin Franke [1,2,15], Rita Falcone[1], Miriam Federer[1,16], Emanuela Ponzi[4], Romina Ficarella[4], Francesca Novara[5], Lars Wittler[6], Bernd Timmermann[7], Mattia Gentile [4], Orsetta Zuffardi[8], Malte Spielmann[9,10,11] & Stefan Mundlos [1,2,12] ✉

Split-Hand/Foot Malformation type 3 (SHFM3) is a congenital limb malformation associated with tandem duplications at the *LBX1*/*FGF8* locus. Yet, the disease patho-mechanism remains unsolved. Here we investigate the functional consequences of SHFM3-associated rearrangements on chromatin conformation and gene expression in vivo in transgenic mice. We show that the *Lbx1*/*Fgf8* locus consists of two separate, but interacting, regulatory domains. Re-engineering of a SHFM3-associated duplication and a newly reported inversion in mice results in restructuring of the chromatin architecture. This leads to ectopic activation of the *Lbx1* and *Btrc* genes in the apical ectodermal ridge (AER) in an *Fgf8*-like pattern induced by AER-specific enhancers of *Fgf8*. We provide evidence that the SHFM3 phenotype is the result of a combinatorial effect on gene misexpression in the developing limb. Our results reveal insights into the molecular mechanism underlying SHFM3 and provide conceptual framework for how genomic rearrangements can cause gene misexpression and disease.

At the sub-megabase scale, chromosomes are organized into distinct regions of high interaction called Topologically Associating Domains (TADs) that are separated from each other by boundaries[1,2]. TADs are thought to restrict enhancer-promoter contacts thereby defining

regulatory domains. Structural variations (SVs), such as deletions, duplications and inversions, can disrupt these functional units and different types of SVs can induce distinct topological changes. For instance, deletions are mainly responsible for TAD fusion, duplications

[1]Max Planck Institute for Molecular Genetics, RG Development & Disease, Berlin 14195, Germany. [2]Institute of Medical and Human Genetics, Charité Universitätsmedizin Berlin, Berlin 10117, Germany. [3]Department of Computational Molecular Biology, Max Planck Institute for Molecular Genetics, Berlin 14195, Germany. [4]Medical Genetics Unit, Department of Reproductive Medicine, ASL Bari, Bari 70131, Italy. [5]Microgenomics Laboratory, Pavia 27100, Italy. [6]Department of Developmental Genetics, Transgenic Unit, Max Planck Institute for Molecular Genetics, Berlin 14195, Germany. [7]Sequencing Core Facility, Max Planck Institute for Molecular Genetics, Berlin 14195, Germany. [8]Department of Molecular Medicine, University of Pavia, Pavia 27100, Italy. [9]Institute of Human Genetics, Universitätsklinikum Schleswig Holstein Campus Kiel and Christian-Albrechts-Universität, Kiel 24118, Germany. [10]Institute of Human Genetics, University of Lübeck, Lübeck, Germany. [11]Human Molecular Genomics Group, Max Planck Institute for Molecular Genetics, Berlin 14195, Germany. [12]Berlin-Brandenburg Center for Regenerative Therapies (BCRT), Charité Universitätsmedizin Berlin, Berlin 13353, Germany. [13]Present address: Department of Pathology, New York University School of Medicine, Langone Health Medical Center, New York, NY 10016, USA. [14]Present address: Kennedy Institute of Rheumatology, University of Oxford, Oxford OX3 7FY, UK. [15]Present address: Centro Andaluz de Biología del Desarrollo, Consejo Superior de Investigaciones Científicas/Universidad Pablo de Olavide, Seville 41013, Spain. [16]Present address: Universität Innsbruck, Innsbruck 6020, Austria. [17]These authors contributed equally: Giulia Cova, Juliane Glaser. ✉e-mail: giulia.cova@nyulangone.org; mundlos@molgen.mpg.de

can promote the formation of a new TAD (neo-TAD), while TAD shuffling is frequently observed upon an inversion[3–5].

SVs can not only affect gene dosage but also modulate basic mechanisms of gene regulation. Indeed, SVs can alter the copy number of regulatory elements or modify the 3D genome by disrupting higher-order chromatin organisation. This can give rise to ectopic enhancer-promoter contacts, gene misregulation and ultimately disease[3–6]. However, the general applicability of this concept remains under investigation, as disruption of TADs and enhancer-promoter interactions are not always accompanied by changes in expression. Studies in Drosophila[7] and in human congenital chromothrypsis cases[8] indicate indeed a degree of robustness of the genome against such events.

SHFM is a congenital limb malformation characterized by variable defects of the central rays of the autopod often together with syndactyly and/or aplasia/hypoplasia of the phalanges, metacarpals, and metatarsals[9]. SHFM is known as a paradigm for genetic heterogeneity. While duplications on either 17p13.3 or 10q24[10] are the most frequent genetic cause, SHFM can be also associated with coding and noncoding mutations at other loci. Indeed, SVs involving the *DYNC1I1* locus where regulatory elements for *DLX5* and *DLX6* are localized[11], and mutations in *TP63, DLX5*, or *WNT10B*[12] have been previously associated with SHFM.

The common underlying patho-mechanism of these mutations is thought to be the failure to maintain a proliferation/growth signal from the apical ectodermal ridge (AER), a signalling center of ectodermal cells at the distal end of the limb bud[13]. SHFM is also a clinically heterogeneous disease[14]. There is a high degree of clinical variability in individuals with SHFM-associated SVs even in the same family[15,16]. In addition, there can be high variability within affected individuals suggesting stochastic and other complex patho-mechanism that are not merely associated with gene dosage alteration.

Tandem duplications on chromosome 10q24 at the *LBX1/FGF8* locus are considered the cause for SHFM type 3 (SHFM3)[15], but the specific patho-mechanism of SHFM3-associated duplications remains unsolved. The reported duplications are at least 500 kb in size, encompassing five genes of the locus (*LBX1, BTRC, POLL, DPCD*, and *FBXW4*) but excluding the neighbouring *FGF8* (Supplementary Fig. 1)[17]. While *LBX1* is essential for early muscle cell differentiation and migration[18], *FGF8* is expressed in the AER together with other FGFs where it functions as a growth factor for the underlying mesechyme[19]. *BTRC* is part of the E3 ubiquitin-protein ligase complex which mediates the ubiquitination and subsequent proteasomal degradation of target proteins[20], *DPCD* is involved in the generation and maintenance of ciliated cells[21], *POLL* encodes for a DNA polymerase[22], whereas *FBXW4* is also involved in ubiquitin-mediated protein degradation[23].

Duplications can have different effects on gene regulation depending on their relative position to chromatin domain boundaries. Studies of the *SOX9* locus, for example, have shown that duplications within the *SOX9* TAD can lead to *SOX9* mis/overexpression and sex reversal, whereas duplications that include the *SOX9/KCNJ2* boundary and the *KCNJ2* gene result in *KCNJ2* misexpression and brachydactyly[4]. In the latter situation, a neo-TAD is formed leading to contact of the *SOX9* enhancers with the *KCNJ2* promoter and *KCNJ2* expression in a *SOX9-like* pattern. Given these examples, it is conceivable that the SHFM3-associated duplications may have specific regulatory effects that could result in gene misexpression.

In addition to the SHFM3 duplications at the *LBX1/FGF8* locus, we reported here a new case of SHFM3 malformation associated with an inversion at the same locus. This inversion was also of particular interest as the patient is a good example of clinical heterogeneity of this limb malformation within the same individual (Supplementary Fig. 2). To uncover the molecular mechanism underlying this congenital disease, we engineered a duplication and the inversion at the *Lbx1/Fgf8* locus in mice using CRISPR/Cas9 genome editing tool[24]. While only the inversion led to a digit phenotype in mice, both SVs

induced similar molecular phenotypes. Chromosome conformation capture Hi-C (cHi-C) analysis of the mouse developing limbs in mutants showed that both SVs led to a disruption of the wild-type three-dimensional chromatin configuration of the locus and ectopic enhancer-promoter contacts. Strikingly, this caused misexpression of two genes, *Lbx1* and *Btrc*, in an *Fgf8*-like pattern in the AER.

Here, we show the impact of SVs at the *Lbx1/Fgf8* locus on chromatin architecture and gene regulation and reveal the complex molecular patho-mechanism underlying SHFM3 limb malformation. With the emergence of novel sequencing technologies and the improvement in the discovery and characterization of SVs, this study illustrates the myriad ways by which SVs can cause disease (i.e., a combination of enhancer repositioning and gene misexpression), highlighting the challenging task of their interpretation in the clinic.

## Results

### SHFM3-associated SVs include TAD boundary and *Fgf8* enhancers

To characterize the 3D conformation at the *Lbx1/Fgf8* locus during mouse development, we performed cHi-C in wild-type murine limb buds at E11.5, a developmental stage where both *Lbx1* and *Fgf8* limb developmental genes are expressed (Fig. 1a). We observed that the locus is characterized by the presence of a major chromatin loop (Fig. 1a, top dashed circle and double dashed lines) comprising two distinct smaller chromatin regulatory domains (Fig. 1a, single dashed lines). The centromeric and the telomeric domains contain the *Lbx1/ Btrc* and the *Fbxw4/Fgf8* genes, respectively, and are hereafter referred to as *Lbx1* and *Fgf8* TADs (Fig. 1a). These two domains are separated from neighbouring TADs and each other by CTCF-associated TAD boundaries in the classical divergent orientation[25] (Fig. 1a). *Poll* and *Dpcd* are located within the boundary region separating *Lbx1* and *Fgf8* TADs and their expression might also contribute to the boundary. Given the divergent roles and expression pattern of *Lbx1* and *Fgf8* during limb development[26,27] (Fig. 1a), separated regulatory domains can be expected to ensure correct gene regulation. The *Fgf8* TAD contains several well characterized enhancers with AER-specific activity as well as other tissue-specific enhancers corresponding to the *Fgf8* expression pattern in the embryo, such as the branchial arches, the somites, and the midbrain-hindbrain boundary[28,29]. Interestingly, all identified AER enhancers[28] are located within a 40 kb region in the introns of *Fbxw4* (58, 59, 61, 66; yellow ovals in Fig. 1a), with the exception of one enhancer (80; orange oval in Fig. 1a) which is located in close proximity to *Fgf8*.

Virtual 4C analysis with viewpoints on *Lbx1* and *Fgf8* further emphasized how the interactions of these genes are to a large extend restricted to their respective regulatory domains (interactions are highlighted in yellow in Fig. 1a). Virtual 4C demonstrates the specific domains of interaction but also shows a domain of overlapping interaction at the TAD boundary region (highlighted in violet in Fig. 1a). Despite being two separate domains, the *Lbx1* and *Fgf8* TADs are also connected via an additional loop (Fig. 1a, top dashed circle in cHi-C map). This interaction is also visible in our 4C analysis (highlighted in pink in Fig. 1a) suggesting that subpopulations of limb bud cells might have different configurations. This suggests a certain degree of leakiness of the *Lbx1/Fgf8* boundary, leading to a level of intermingling of the two domains despite the very different expression patterns of *Lbx1* and *Fgf8*.

Mapping of in-house and already published SHFM3 duplications onto the 3D structure of the locus showed that they all include the boundary between the *Lbx1* and *Fgf8* TADs (Fig. 1b). Furthermore, all duplications comprised *Lbx1* at the centromeric side and the previously identified *Fgf8* AER enhancers located within *Fbxw4*[29], while excluding *Fgf8* at the telomeric side (Fig. 1b and Supplementary Fig. 1). Mapping of the new SHFM-associated inversion showed that the boundary region between the *Lbx1* and *Fgf8* TADs and the *Fgf8* AER

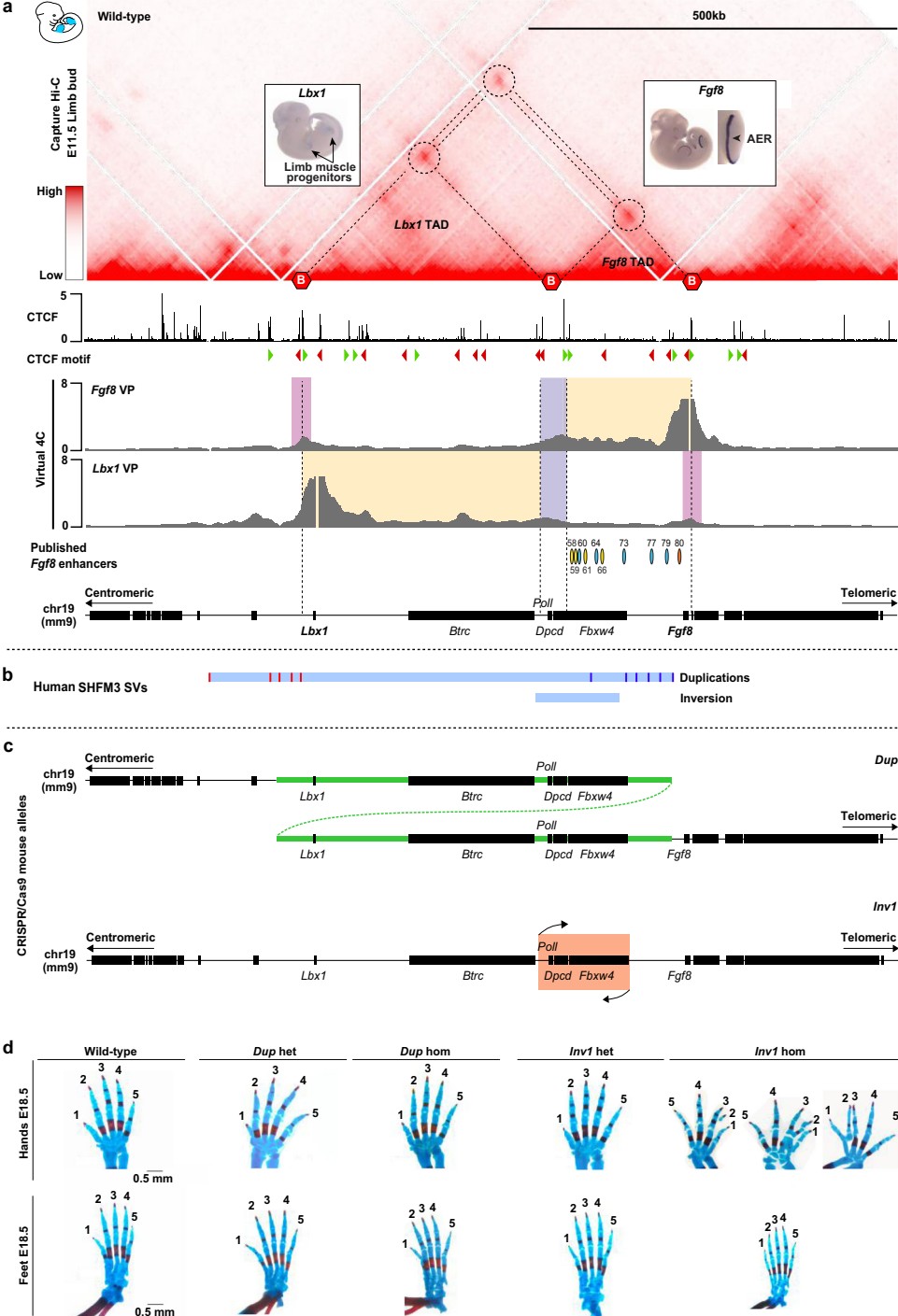

**Fig. 1 | Engineering of the SHFM-associated structural variations at the Lbx1/Fgf8 locus. a** cHi-C (data are shown as merged signal of *n* = 3 biological and 1 technical replicates) of extended *Lbx1/Fgf8* locus generated from wild-type E11.5 mouse limb buds. *Lbx1* and *Fgf8* are located within distinct TADs (indicated by single dashed lines) separated by boundaries (indicated by red hexagons) in correlation with CTCF binding sites, as indicated in ChIP-seq from E11.5 mouse limb buds[61] below. Loops (dashed circles) indicate interaction between CTCF sites. Interaction plots using virtual 4C from *Fgf8* and *Lbx1* viewpoints (VP) are shown below. Regions of interactions relative to *Lbx1* and *Fgf8* within their own TADs are highlighted in yellow, whereas contacts with the boundary region between *Lbx1* and *Fgf8* TADs and those over such boundary are in violet and pink, respectively. Published *Fgf8* enhancers[28] are indicated by ovals. Yellow ovals highlight enhancers driving *Fgf8* expression in the AER and localized within the introns of *Fbxw4*, while in orange is the only AER enhancer in close proximity to *Fgf8*. **b** Schematic of human SHFM3 related structural variations (SVs). Red and blue lines in the duplications bar represent the different centromeric and telomeric breakpoints, respectively. Breakpoints of the inversion are shown below. **c** Schematic of the SHFM-associated SVs engineered in mice using CRISPR/Cas9 genome editing tool, particularly one selected tandem duplication (*Dup*) and the inversion (*Inv1*). **d** Skeletal analysis of fore- and hindlimb stained with alcian blue (cartilage) and alizarin red (bone), from wild-type, heterozygous and homozygous *Dup* and *Inv1* 18.5 embryos. No particular phenotype was observed in both heterozygous and homozygous *Dup* and in heterozygous *Inv1*, whereas fused bones and split digits were detected in homozygous *Inv1* forelimbs.

enhancers are located within the inversion breakpoints, suggesting a potential common patho-mechanism.

To investigate the effect of the SHFM3-associated duplications and inversion on chromatin configuration, gene expression, and phenotype, we engineered a human SHFM3 duplication (*Dup*) and the newly reported inversion (*Inv1*) in mice (Fig. 1c).

### *Inv1* but not *Dup* results in a SHFM-like phenotype

First, we analysed the mutant mice for skeletal phenotypes at developmental stage E18.5. In both heterozygous and homozygous *Dup* and in heterozygous *Inv1* we did not observe any skeletal malformations in the extremities of all the analysed embryos (n = 4 for *Dup* het, n = 5 for *Dup* hom and n = 8 for *Inv1* het) (Fig. 1d). Furthermore, none of the >25 adult mice with those genotypes from the established *Dup* and *Inv1* mouse lines displayed a visible morphological phenotype. However, 3 out of 11 E18.5 homozygous *Inv1* embryos exhibited skeletal defects with a tendency to digit separation and partially missing or fused digits/bones (Fig. 1d), resembling the phenotype of SHFM patients[15] and particularly that of the SHFM3 inversion individual (Supplementary Fig. 2). Additionally, all homozygous *Inv1* mutants showed underdeveloped forelimbs and hindlimbs (Supplementary Fig. 3a). This latter phenotype is highly similar to the *Fgf8* conditional knockout[27,30], suggesting that the expression of *Fgf8* in the developing limb might be affected in the *Inv1* homozygous mutants. Loss of *Fgf8* expression is known to affect the proper development of humerus, radius, ulna, and to cause loss mainly of the first digit[27], an overall phenotype different from SHFM3 based on the clinical cases reported so far in literature[9,15,17].

### SHFM3-associated SVs result in ectopic 3D interactions

The phenotype observed in the homozygous *Inv1* mutants and the phenotypic discrepancy between the two types of SHFM3 SVs prompted us to study the effect on 3D genome organization during limb development. We thus performed cHi-C in homozygous *Dup* and *Inv1* E11.5 murine limb buds (Fig. 2). Compared to wild-type (Fig. 2a), the cHi-C map of the *Dup* mutant (Fig. 2c) showed a general increase in the frequency of interactions within the duplicated region, reflecting the copy number increase. In the subtraction map (*Dup* versus wild-type), the effect of the higher copy number on the signal intensity is corrected (Fig. 2c, mirror view). Interestingly, we observed a stripe of increased interactions (indicated by black arrowheads) involving the part of the *Fgf8* TAD containing the *Fgf8* AER enhancers comprised in the duplication (blue bar in Fig. 2c) and a region immediately upstream of the *Lbx1* centromeric TAD boundary (orange bar in Fig. 2c). This reflected the formation of a neo-TAD, as illustrated in the schematic of the predicted linear configuration (Fig. 2d). We also observed increased contacts between the entire *Lbx1* TAD and the *Fgf8* AER enhancers region (indicated by dashed rectangle and asterisks in Fig. 2c). These interactions were established by the *Fgf8* AER enhancers from the neo-TAD, with the endogenous and/or the duplicated *Lbx1* TADs (Fig. 2d). Indeed, in the *Dup* mutant, interactions between the *Fgf8* AER enhancers and the telomeric duplicated *Lbx1* TAD is favoured by a shorter distance, particularly with *Lbx1* gene (-130 kb in *Dup* versus -380 kb in wild-type configuration), and by the presence of a weaker TAD boundary (two CTCF binding sites instead of four). Moreover, the *Fgf8* AER enhancers could also contact the centromeric endogenous *Lbx1* TAD due to the previously mentioned leakiness of the boundary and the absence of their target gene *Fgf8*, making these enhancers free to interact with the neighbouring *Lbx1* gene and the entire *Lbx1* TAD.

Ectopic interactions between the *Fgf8* AER enhancers region and the *Lbx1* TAD were also observed for the *Inv1* mutant (Fig. 2e,f). In the inversion, the boundary between *Lbx1* and *Fgf8* TADs functioned as an anchor point around which part of the *Fgf8* TAD was moved into the *Lbx1* TAD and vice versa. Since the centromeric breakpoint of the

inversion is relatively close to the boundary, the *Lbx1* TAD was enlarged while the *Fgf8* TAD was reduced in size. As a consequence, the *Fgf8* AER enhancers were now relocated in the *Lbx1* TAD resulting in increased contacts (dashed rectangle and black asterisks in Fig. 2e) with the repositioned *Fgf8* AER enhancer region, similar to our finding in the *Dup*. As a consequence of the reconfiguration of the locus, *Fgf8* was now isolated from its AER enhancers and we indeed observed a loss of interactions between these elements and their endogenous gene (dashed rectangle with white asterisk in Fig. 2e). Since the SHFM3 phenotype in humans manifests already in heterozygosity, we analysed the 3D chromatin configuration in *Dup* and *Inv1* het embryos and detected the same ectopic interactions as in homozygosity, but with lower frequency (Supplementary Fig. 3b, c).

Given that the overall configuration of the locus with two main TADs is conserved in humans (Fig. 3a), we then investigated whether the observed interaction changes were also present in SHFM3 patients. We performed 4C-seq experiments in human fibroblasts from a patient carrying the SHFM3 duplication (fibroblasts from the inversion case were not available) and exhibiting the classical SHFM phenotype. We detected an increased interaction between the 125 kb region of *FGF8* TAD containing the AER enhancers and both *LBX1* and *BTRC* genes compared to healthy control (Fig. 3b). This effect appeared to be specific as the 4C signal remained unchanged when using the viewpoint of *FGF8* or when focusing on a region outside of the *FGF8* TAD (Fig. 3b and Supplementary Fig. 4a). As we showed that mouse virtual 4C is a good way of similarly quantifying cHi-C data (Fig. 1a), we used it for the *Dup* and the *Inv1* mouse mutants. We confirmed that the same ectopic interactions as in the patient data existed in the *Dup* mutants (Fig. 3c, d and Supplementary Fig. 4b), indicating the applicability of our mutant mouse model for SHFM3. While a similar interaction between *Lbx1/Btrc* and the 125 kb AER enhancers region was present in the *Inv1* mutant, we also observed ectopic interactions between *Lbx1/Btrc* and the *Fgf8* gene that were specific to this mutant (Fig. 3c, d and Supplementary Fig. 4b). We also detected a decreased interaction between *Fgf8* promoter and its enhancer region, again suggesting *Fgf8* expression might be affected in the *Inv1* mutant (Fig. 3c, d).

### *Dup* and *Inv1* lead to *Lbx1* and *Btrc* misexpression in the AER

The 3D conformation data were not sufficient to explain the discrepancy of morphological phenotype between the *Inv1* and *Dup* mutants. To investigate the effect of the observed chromatin rearrangements on gene expression and possibly link it to the observed phenotype, we performed detailed expression analysis (RNA-seq, single-cell RNA-seq and whole mount in situ hybridisation (WISH)) at developmental stage E11.5. In the *Dup* hom mutant, all genes within the duplication (*Lbx1*, *Btrc*, *Poll*, *Dpcd* and *Fbxw4*) were significantly upregulated in the developing limb buds, corresponding to the copy number increase (Fig. 4a and Supplementary Data 1). In the *Inv1* hom mutant, *Fbxw4* and *Fgf8* were downregulated, reflecting the disruption of the *Fbxw4* and the repositioning of the *Fgf8* AER enhancers to the neighboring *Lbx1* TAD (Fig. 4a and Supplementary Data 1). Strikingly, *Lbx1* and *Btrc* exhibited increased expression levels in the *Inv1* mutant, indicating that the observed chromatin rearrangement had an effect on gene expression. Genes outside of the *Fgf8* and *Lbx1* TADs (*Sif2*, *Twnk* and *Oga*) remained unchanged in both *Dup* and *Inv1* mutants (Fig. 4a).

To investigate whether these changes of gene expression were specific to certain regions of the limb bud, we performed single-cell RNA-seq (scRNA-seq) from E11.5 wild-type, *Dup* hom and *Inv1* hom limb buds. Individual clusters corresponding to the major cell types in the developing limb were identified including mesenchyme, dorso-ventral ectoderm, AER and muscle as well as satellite cell types such as neurons, lymphocytes, keratinocytes and endothelial cells (Fig. 4b and Supplementary Fig. 5a–c). In wild-type limbs, *Fgf8* expression was only present in the AER and *Lbx1* was expressed only in muscle cells whereas

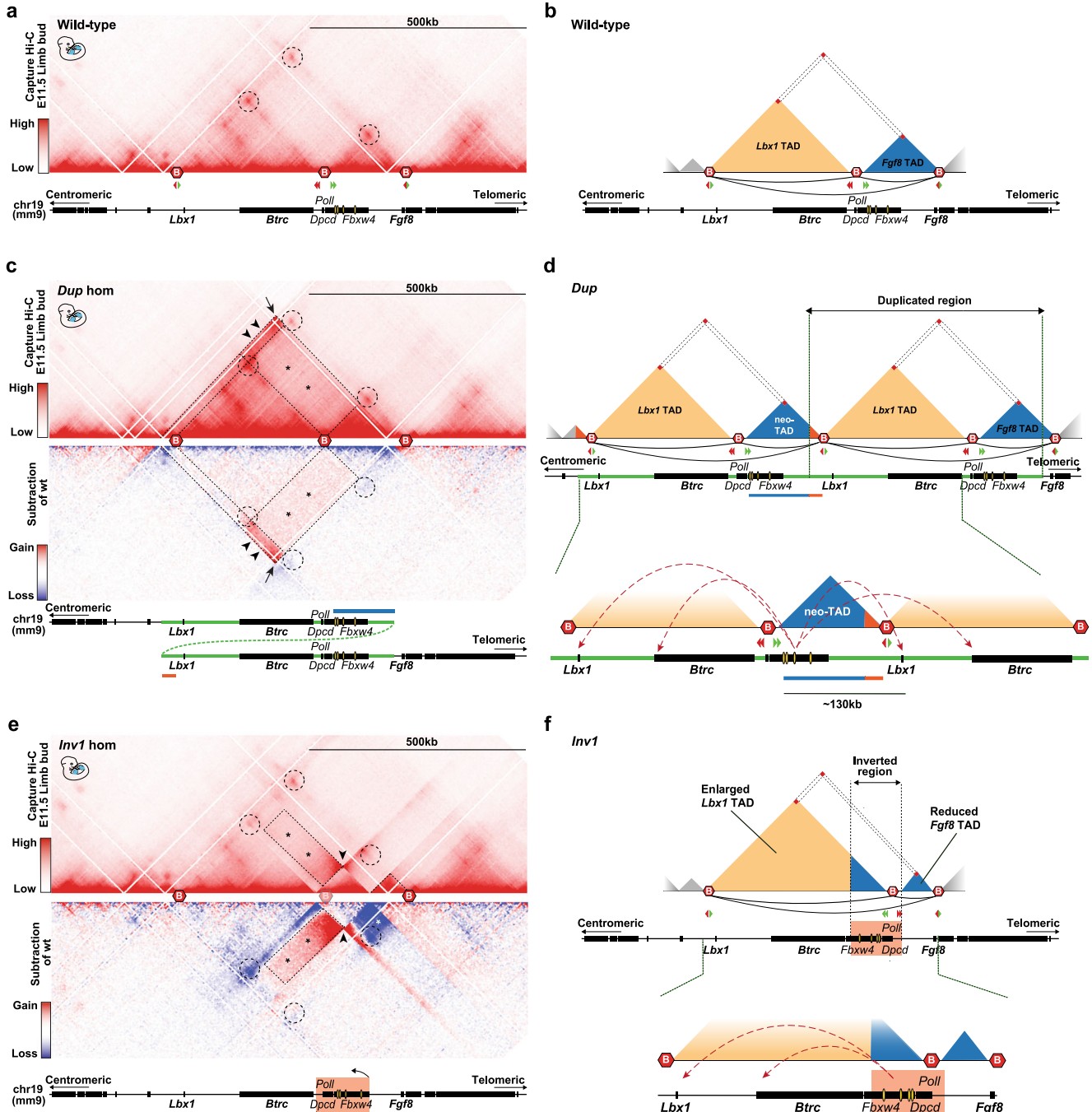

**Fig. 2 | Ectopic 3D interactions upon structural rearrangements at the *Lbx1/Fgf8* locus. a** Wild-type E11.5 mouse limb buds cHi-C (merged signal of *n* = 3 biological and 1 technical replicates). Red hexagons indicate boundaries, dashed circles highlight boundaries interactions. CTCF binding sites orientation and schematic of the locus are below. Yellow ovals highlight *Fgf8* AER enhancers. **b** Schematic of the wild-type configuration. *Lbx1* TAD in yellow, *Fgf8* TAD in blue. Red squares highlight boundaries interactions. CTCF binding sites orientation, interactions between convergent CTCF and schematic of the locus are below. **c** cHi-C (merged signal of *n* = 3 biological and 1 technical replicates) of homozygous *Dup* E11.5 mouse limb buds. Subtraction map between wild-type and *Dup* interactions is shown in mirror view. Dashed circles indicate positions of preserved wild-type boundaries interactions, black arrow highlight the new contact reflecting the duplication breakpoints. Black arrowheads indicate gain of contacts between the region containing the *Fgf8* AER enhancers (blue bar in the locus schematic below) and the region flanking the centromeric side of the *Lbx1* TAD boundary (orange bar). Asterisks highlight the rectangular dashed area showing increased interactions

between the *Fgf8* AER enhancers region and the *Lbx1* TAD. **d** Schematic of the predicted linear *Dup* configuration. The entire *Lbx1* TAD is duplicated, remaining unchanged in its content and configuration, like the telomeric *Fgf8* TAD. Between the two *Lbx1* TADs, a neo-TAD forms. Below is a zoom-in view with red dashed arrows indicating potential ectopic contacts. **e** cHi-C (merged signal of *n* = 3 biological and 1 technical replicates) of homozygous *Inv1* E11.5 mouse limb buds. Dashed circles indicate the positions of the original wild-type boundaries interactions, two (centromeric and telomeric) lost upon inversion. The inverted boundary is shown as blurry. Dashed lines on the right point out the new smaller *Fgf8* TAD. Black arrowhead highlights the bow tie configuration representative of inverted regions. Black asterisks highlight the rectangular dashed area showing ectopic interactions between the *Fgf8* AER enhancers region and the *Lbx1* TAD, white asterisk points out the loss of interactions between *Fgf8* and its AER enhancers. **f** Schematic of the *Inv1* configuration with zoom-in of the locus highlighting potential ectopic interactions (red dashed arrows).

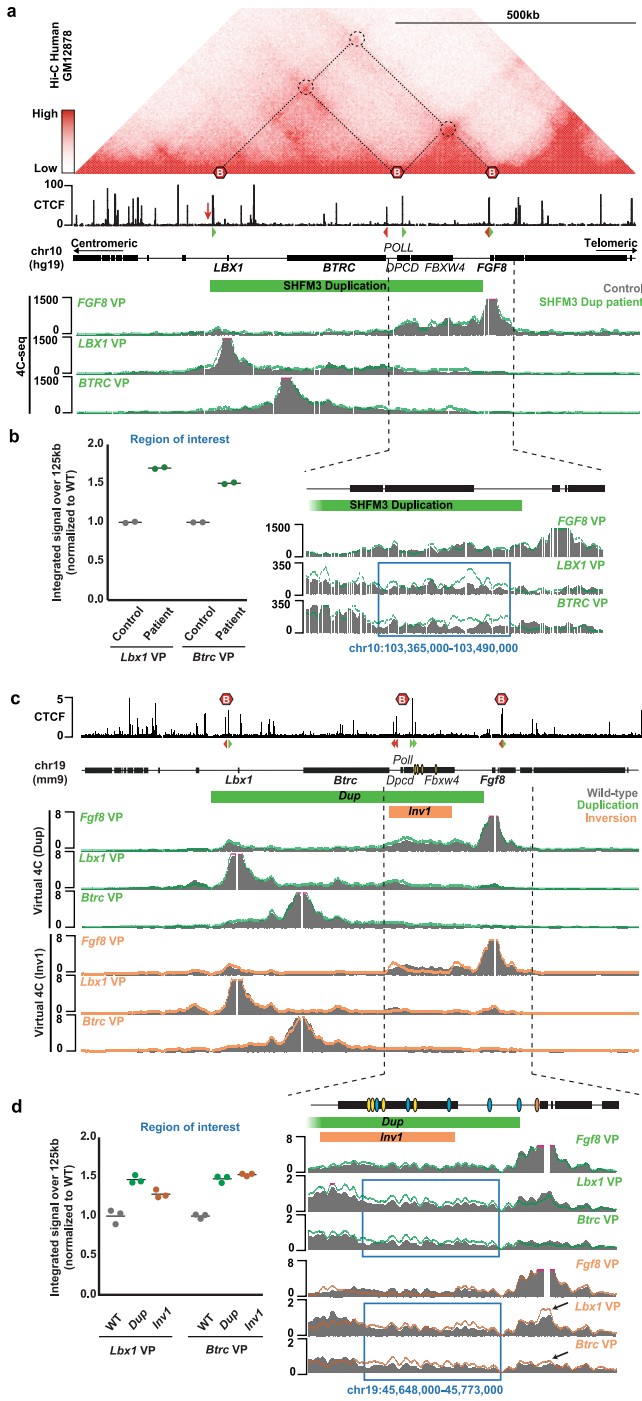

**Fig. 3 | 4C-seq in SHFM3 patient fibroblasts reveals ectopic interactions involving *LBX1* and *BTRC*, further supported by virtual 4C in mouse. a** Hi-C at the *LBX1/FGF8* locus (hg19; chr10:102,668,128-103,840,922) derived from GM12878 cell[50]. TADs structures are similar between human and mouse. CTCF ChIP-seq from GM12878 cells[62,63], orientation of CTCF binding sites and schematic of the human *LBX1/FGF8* locus are shown below. Red arrow indicates a CTCF binding site in antisense orientation at the *LBX1* centromeric TAD boundary present in mouse but not in human. Below is the normalized 4C-seq of human fibroblasts healthy control (grey) versus a SHFM3 patient carrying the duplication (green line) with viewpoints (VP) on *FGF8*, *LBX1* and *BTRC* promoters. Data are shown as merged signal of *n* = 2 replicates. **b** (Left panel) 4C-seq integrated signal over a 125 kb region comprising the *FGF8* regulatory elements that are duplicated. (Right panel) A zoom-in view from the 4C-seq showing the region used for the integrated signal (blue rectangle). We observed an increase of contacts between both *LBX1* and *BTRC* promoters and the *FGF8* regulatory elements which is not present in control regions (Supplementary Fig. 4). **c** Similar representation as in (**a**) showing the mouse CTCF and virtual 4C (generated from E11.5 limbs cHi-C maps from Fig. 2) tracks. Wild-type (WT) signal is shown in grey and *Dup* and *Inv1* with a green and orange line, respectively. Data are shown as merged signal of *n* = 3 replicates. **d** (Left panel) Virtual-4C integrated signal over the 125 kb region comprising *Fgf8* regulatory elements that are affected by the duplication and the inversion. (Right panel) A zoom-in view from the virtual-4C showing the 125 kb used for the integrated signal (blue rectangles). Black arrow highlights a gain of contact observed only for *Lbx1* view point with the *Fgf8* gene, outside of the region of interest. Gain of interactions is observed between both *Lbx1* and *Btrc* promoters and the *Fgf8* regulatory elements which is not present in control regions (Supplementary Fig. 4).

and *Btrc* were activated by the *Fgf8* AER enhancers now able to contact the two genes' promoters (Fig. 3d). Thus, the scRNA-seq data confirmed the bulk RNA-seq data and showed, in addition, that *Lbx1* and *Btrc* were misexpressed in the AER with a stronger effect in the *Inv1*.

Next, we performed WISH to confirm and further investigate potentially altered expression patterns. We confirmed that *Lbx1*, normally expressed in migrating muscle cells (black arrowheads for *Lbx1* in Fig. 5a), and *Btrc* were ectopically expressed in the AER, in both homozygous (Fig. 5a) and heterozygous (Supplementary Fig. 6a) *Dup* and *Inv1* E11.5 embryos. As observed in the scRNA-seq data, an even more pronounced misexpression of both genes was also observed in the *Inv1* hom mutants (Fig. 5a). In the *Dup* mutant, *Fgf8* expression was unchanged compared to wild-type. In contrast, in the *Inv1* mutant we confirmed a complete loss of *Fgf8* expression in homozygosity (Fig. 5a) and a partial loss in heterozygosity (Supplementary Fig. 6a), likely due to the loss of the endogenous interactions of *Fgf8* with its enhancers. Finally, we did not detect any misexpression in the AER for *Poll*, *Dpcd* and *Fbxw4*, neither in *Dup*, nor in the *Inv1* mutants (Supplementary Fig. 6b).

Our findings suggest that the four previously described *Fgf8* AER enhancers are the main regulators of *Fgf8* expression in this tissue[28,29]. To confirm that, we used CRISPR/Cas9 technology to delete those four enhancers (CE58, CE59, CE61 and CE66) in a wild-type background (Δ*Fgf8*-AER-enh, Supplementary Fig. 7a) and compared the outcome to what we observed in the *Inv1* E11.5 embryos. *Fgf8* expression appeared to be strongly affected in both mutants (Supplementary Fig. 7b–c), resulting in a similar reduction of limb bud size (Supplementary Fig. 7c). *Lbx1* and *Btrc* expression was not affected in the Δ*Fgf8*-AER-enh mutant.

Knowing that these enhancers control the majority of *Fgf8* expression in the AER, we tested whether the repositioning of the AER enhancers was sufficient to induce ectopic expression of *Btrc* and *Lbx1* in the AER. We used CRISPR/Cas9 technology to insert the four *Fgf8* AER enhancers in the *Lbx1* TAD in a wild-type background (*Fgf8 AER enh KI*, Fig. 5b). Given the presence of a CTCF binding site within enhancer CE58 and our aim to not create any new boundary upon knock-in we removed the CTCF recognition motif from enhancer CE58. WISH of *Fgf8 AER-enhancer KI* E11.5 mutant embryos showed

*Btrc* was expressed at a low level in all limb cells with higher expression in the AER (Fig. 4c). The three other genes of the locus were expressed in all major cell types of the developing limb at a rather low level (Fig. 4c). The single-cell data were in agreement with the bulk RNA-seq such that the five genes included in the duplication (*Lbx1, Btrc, Poll, Dpcd and Fbxw4*), but not *Fgf8*, displayed increased expression in the *Dup* mutant compared to wild-type (Fig. 4c). The loss of *Fgf8* expression in the *Inv1* mutant was also confirmed. Interestingly, in the AER, both *Lbx1* and *Btrc* showed a stronger increase of expression in the *Dup* mutant compared to wild-type than what we would expect with the increased copy number, indicating misexpression in the AER. In the *Inv1* mutant, where no copy number changes occurred, we observed the misexpression to be even stronger (Fig. 4c and Supplementary Fig. 5d), particularly for *Btrc*. These results suggested that both *Lbx1*

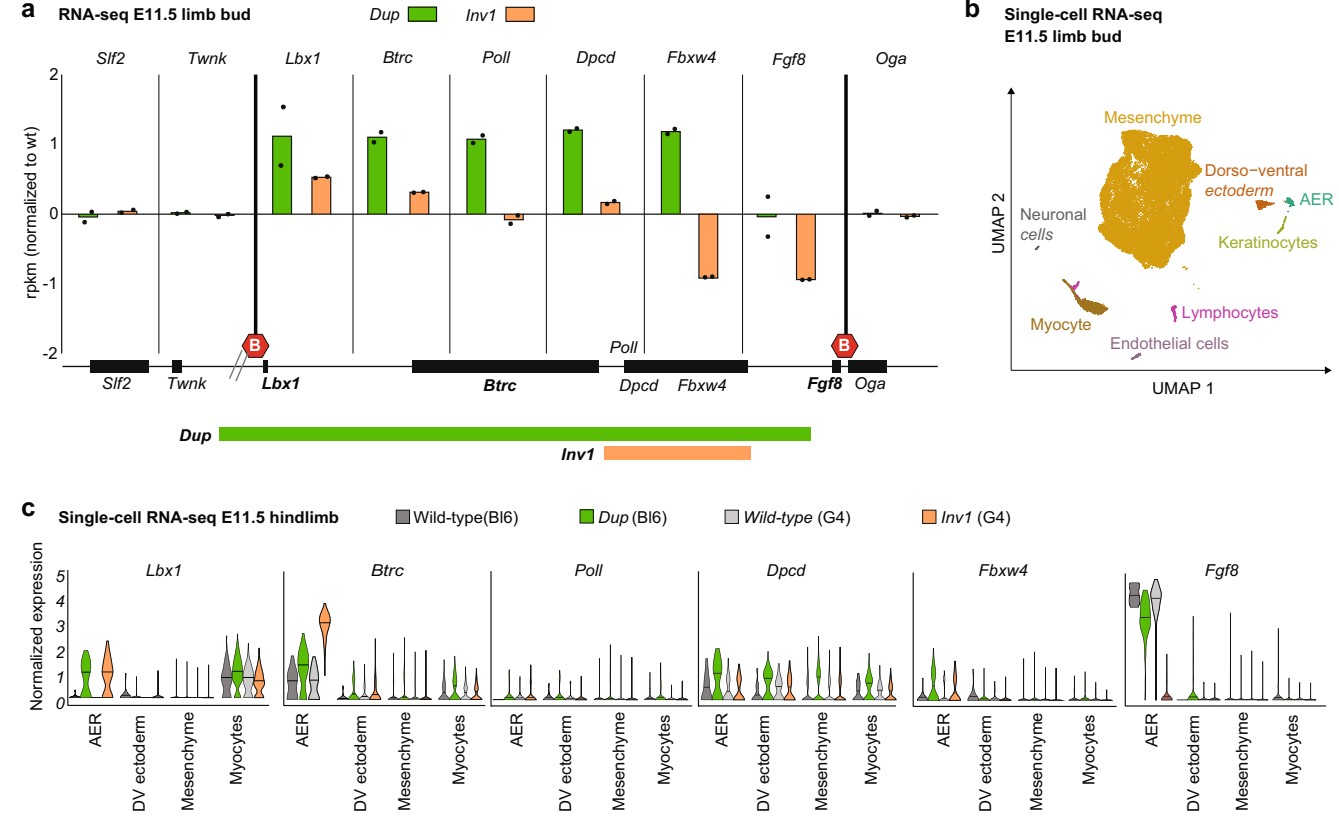

**Fig. 4 | *Dup* and *Inv1* chromatin rearrangements have an impact on gene expression. a** Bar plot representing the rpkm value normalized to wild-type for genes in the *Lbx1* and *Fgf8* TADs and flanking genes (*Slf2* and *Twnk* (centromeric), *Oga* (telomeric)) from RNA-seq data of homozygous *Dup* and *Inv1* E11.5 limb buds. *n* = 2 biological replicates per condition. Schematic of the locus shows the position of the genes, the TAD boundaries (red hexagons) and the parts included in the Dup and the Inv1 mutants. Wild-type *Dup* and *Inv1* samples are shown in black, green and orange, respectively. **b** Uniform manifold approximation and projection (UMAP)

showing 8 cell clusters identified via scRNA-seq of E11.5 mouse limbs from wild-type, *Dup* and *Inv1* mutants (*n* = 1). **c** Violin plot representing the normalized expression of the 6 genes at the locus in AER, dorso-ventral (DV) ectoderm, mesenchyme and myocytes from E11.5 hindlimbs. The *Dup* mutant (green) was generated in a C57Bl6 background and thus compared to a C57Bl6 wild-type sample (dark grey) whereas the *Inv1* mutant (orange) was generated in a G4 (129sv x C57Bl6) and compared to a G4 wild-type sample (light grey).

misexpression of *Lbx1* and *Btrc* in the AER (Fig. 5a), supporting the ability of the *Fgf8* enhancers to activate other genes. Of note, we observed that *Lbx1* was ectopically expressed in the AER of all analysed embryos, while *Btrc* was present in the AER of 50% of the examined embryos and displayed a much lower signal. Importantly, this result further confirmed that the observed ectopic expression of *Lbx1* and *Btrc* in the AER in the *Dup* and *Inv1* mutants was due to activation by *Fgf8* enhancers.

### *Lbx1* AER misexpression is associated with myogenic signature
We and others have previously shown that ectopic gene expression during organ formation can lead to developmental malformation[3,4,31–33]. However, the molecular consequences of the observed misexpression are still unknown. In both mutants, *Lbx1* is part of the top 3 up-regulated genes in the AER (Supplementary Data 2). We reasoned that *Lbx1* could activate its gene regulatory network in the AER and to investigate this we performed a Gene Ontology (GO) search using the 200 genes most differentially up-regulated in the AER cells in *Dup* and *Inv1* compared to wild-type (C57Bl6 and G4, respectively) hindlimb (Supplementary Data 2). Strikingly, using the MGI mammalian phenotype terms, 4 out of the 5 and 3 out of the 10 most enriched terms in the *Dup* and *Inv1* comparison respectively, were related to myogenic phenotypes (Fig. 5c and Supplementary Fig. 6c). Genes associated with these terms included *Lbx1*, *Myog*, *Gab1*, *Rela*, *Mapk14* and *Dnm2* which were all expressed at a higher level in more cells in the AER of the *Dup* mutant

compared to wild-type (Fig. 5d). Additionally, expression of one of the genes, *Myog*, was also significantly increased in the bulk RNA-seq differential analysis data from limb buds from the *Inv1* mutant (Supplementary Data 1). Of note, *Lbx1* expression was detectable in almost all AER cells while none of the wild-type AER cells expressed *Lbx1*, corroborating our previous observations. Overall, it suggested that gene misexpression could trigger a specific pathway. Regarding the effect of *Btrc* misexpression, despite this gene being strongly misexpressed in the AER in *Inv1* mutant, the downstream effect of an E3 ubiquitin ligase would be expected to be seen at the protein level and unlikely at the mRNA level[34].

### SHFM-like phenotype requires *Lbx1* and *Btrc* AER misexpression
To further investigate the contribution of *Lbx1* and *Btrc* to SHFM, we engineered two more CRISPR/Cas9 alleles to promote *Lbx1* or *Btrc* misexpression alone in the AER and compared it to *Dup* and *Inv1* where genes were simultaneously misexpressed. First, we generated a second, slightly larger inversion (*Inv2*) including the *Fgf8* AER enhancers and *Btrc* (Fig. 6a). This mutant showed ectopic interactions between the *Fgf8* AER enhancers and *Lbx1* only, as observed in cHi-C and particularly in the subtraction cHi-C map (Fig. 6b) but in general with much lower intensity compared to *Inv1* mutant. Misexpression of only *Lbx1* but not *Btrc* was indeed detected in the AER (Fig. 6D). We also noticed that the embryos exhibited underdeveloped head structure, most likely as a consequence of the midbrain-hindbrain boundary *Fgf8* enhancers[28,29] (CE64 and CE79, see Fig. 1a) being repositioned outside

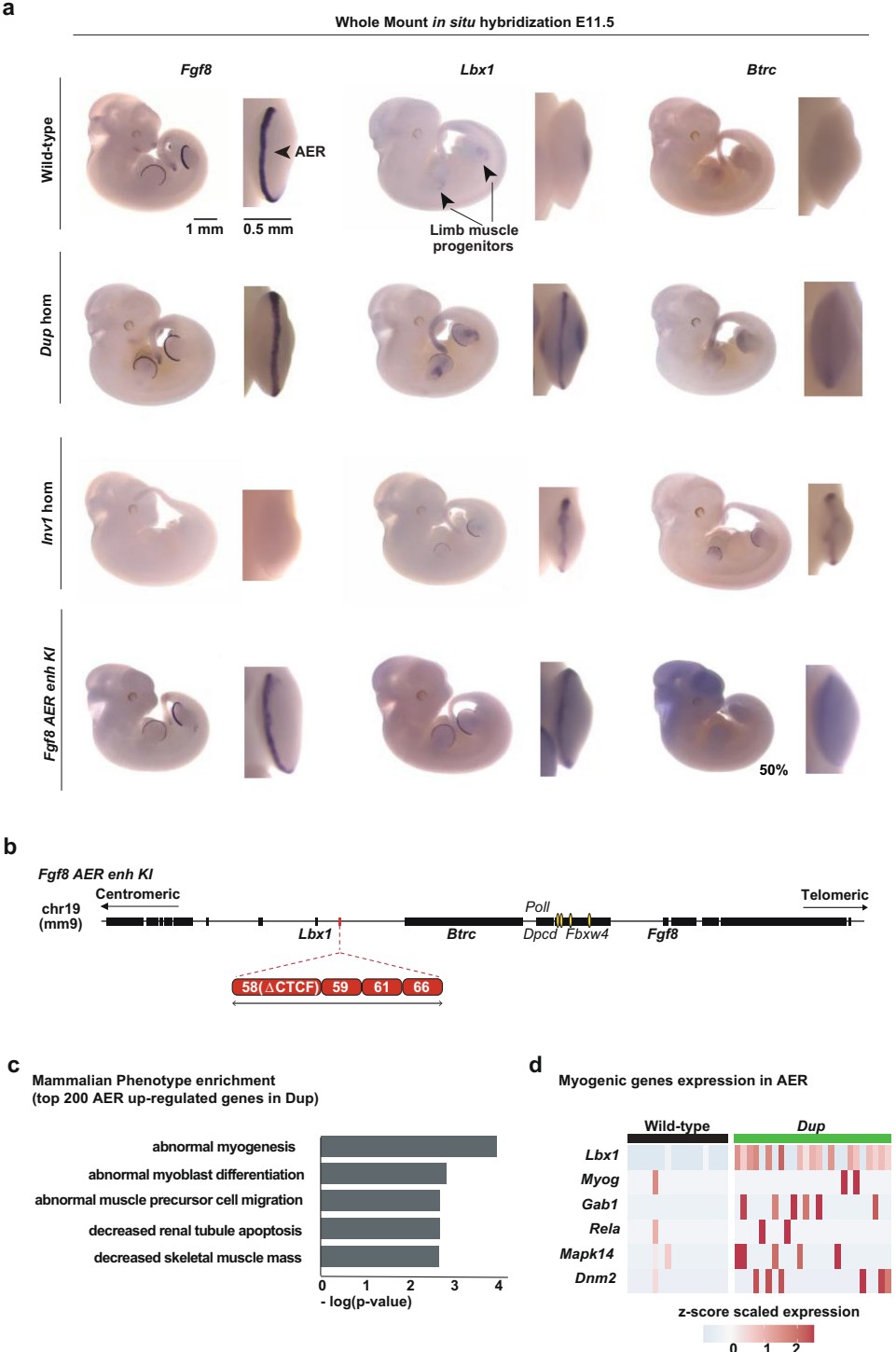

**Fig. 5 | Ectopic interactions result in *Lbx1* and *Btrc* misexpression in the AER. a** Whole-mount in situ hybridization for *Fgf8*, *Lbx1* and *Btrc* at E11.5. Expression was checked and confirmed in at least 3 or more homozygous embryos (at least *n* = 3 biological replicates). **b** Schematic of the *Lbx1/Fgf8* locus for CRISPR/Cas9 *Fgf8 AER enhancer KI* allele. Yellow ovals highlight *Fgf8* AER enhancers. **c** GO analysis for the MGI Mammalian phenotype enrichment terms[59] using the top 200 genes that are up-regulated genes in the *Dup* mutant compared to wild-type in the AER cells from the scRNA-seq data. The 5 most significant enriched terms are represented on a -log10 (*p*-value) scale. *P*-value was computed using a Fisher exact test. **d** Heatmap showing the expression of 6 genes (*Lbx1, Myog, Gab1, Rela, Mapk14* and *Dnm2*) associated with the myogenic enriched terms in (**c**) in the AER cells. Each column represents one AER cell and the z-score scaled expression is indicated for each gene, showing a global increased expression in the *Dup* mutant's AER cells.

of the *Fgf8* TAD in *Inv2*. To investigate the effect of an ectopic expression of *Btrc* alone, we re-targeted the *Inv1* allele and deleted *Lbx1* (*Inv1 ΔLbx1*) (Fig. 6a). As expected, no misexpression for *Lbx1* was detected in the AER and in the limb in general, while *Btrc* was transcriptionally active (Fig. 6d). In addition, we observed a weak expression of *Fgf8* rescued in the AER, underlining the ability of the *Fgf8* AER enhancers to re-contact their endogenous target gene in the absence of *Lbx1* and, similar to the situation in the *Dup* mutant. Skeletal analysis of *Inv2* and *Inv1 ΔLbx1* homozygous mutants (Fig. 6e) revealed underdeveloped limbs due to a complete (*Inv2*) or strong (*Inv1 ΔLbx1*)

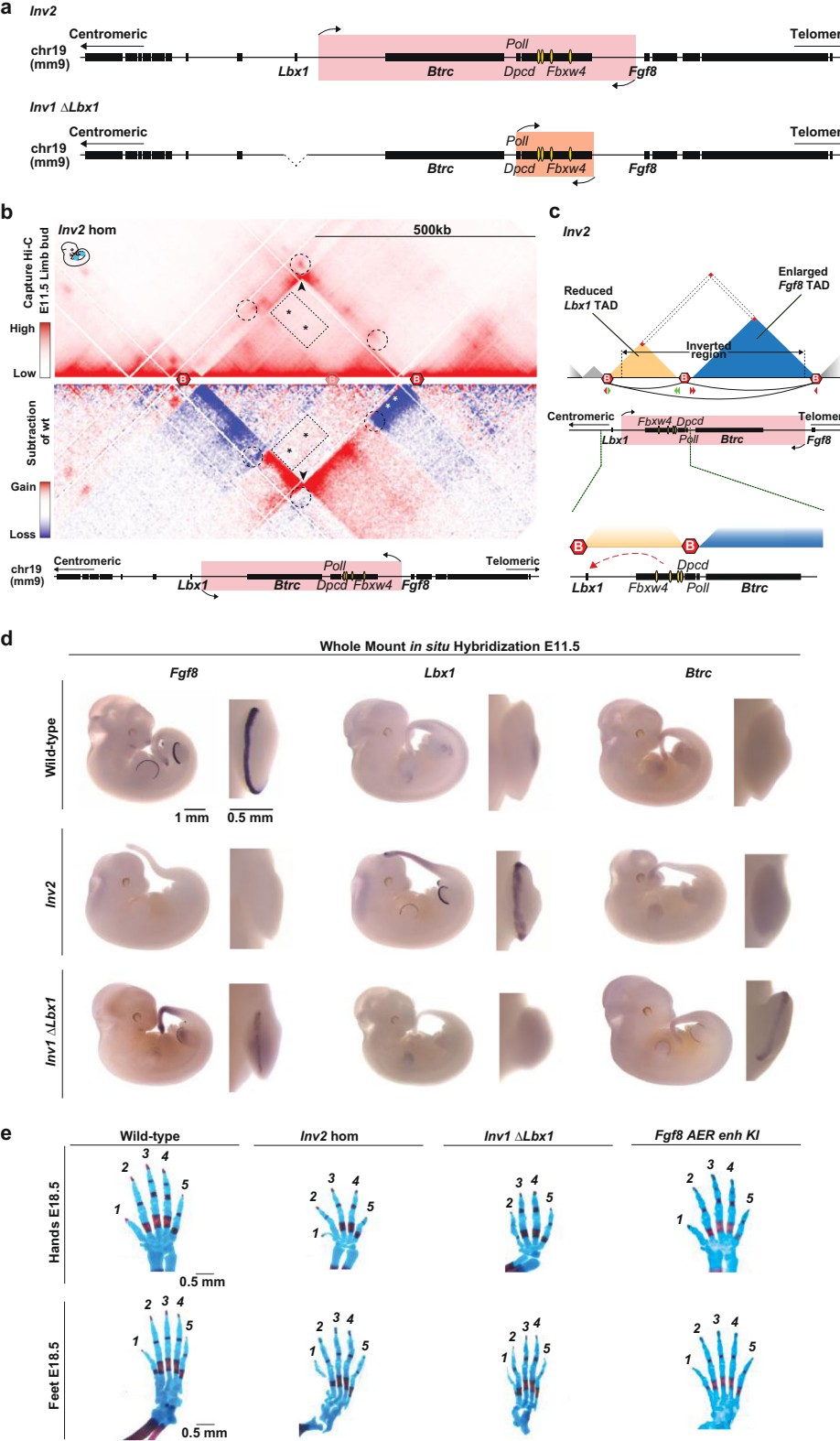

**d** | Whole Mount *in situ* Hybridization E11.5

loss of *Fgf8* expression (Fig. 6d), but none of them (*n* = 10 for *Inv2* and *n* = 7 for *Inv1 ΔLbx1*) showed the level of SHFM phenotype observed in *Inv1* (Fig. 1d), supporting the notion that misexpression of both *Lbx1* and *Btrc* is required to develop the SHFM-like phenotype. Furthermore, the comparison of *Inv1*, *Inv2* and *Inv1 ΔLbx1* indicated that the level of *Fgf8* expression might have a role in triggering the SHFM-like phenotype in combination with *Lbx1* and *Btrc* misexpression. The *Inv2* phenotype, where *Fgf8* expression was completely lost, seemed

indeed more severe than *Inv1 ΔLbx1*, where *Fgf8* was only partially lost (Fig. 6e and Fig. 7).

Finally, the level of *Lbx1* and/or *Btrc* misexpression is likely to also play a role in the manifestation of the limb phenotype. Indeed, in the *Dup* and *Fgf8 AER-enhancer KI* mutants, the level of *Btrc* misexpression was lower than in the *Inv1* mutant (Fig. 5a) and no phenotype was detected (Fig. 1d and Fig. 6e) despite misexpression of both *Lbx1* and *Btrc* genes in the AER.

**Fig. 6 | Misexpression of both *Lbx1* and *Btrc* is required to develop a SHFM phenotype. a** Schematics of the *Lbx1/Fgf8* locus for CRISPR/Cas9 *Inv2* and *Inv1 ΔLbx1* alleles. Yellow ovals highlight *Fgf8* AER enhancers. **b** cHi-C (data are shown as merged signal of *n* = 3 biological and 1 technical replicates) of homozygous *Inv2* from E11.5 mouse limb buds. The subtraction map between wild-type and *Inv2* interactions is shown in mirror view. Dashed circles indicate the position of the original wild-type interactions between boundaries, two of them lost upon reshuffling of the boundary between *Lbx1* and *Fgf8* TADs as a consequence of this inversion. The boundary involved in the inversion is shown as blurry. Black arrowhead highlights the bow tie configuration representative of the inverted regions. The rectangular dashed area highlighted by black asterisks show ectopic interactions compared to wild-type between the *Fgf8* AER enhancers region and the

one containing now only *Lbx1* and not anymore *Btrc*. White asterisks point out the loss of interactions between *Fgf8* and its AER enhancers. **c** Schematic of *Inv2* configuration. Upon this inversion, the *Fgf8* AER enhancers are repositioned within a smaller *Lbx1* TAD, whereas the *Fgf8* TAD is increased in size due to the inversion of the boundary and contains now *Btrc*, isolated from the *Fgf8* AER enhancers. Below is a zoom-in of the locus highlighting the potential ectopic interactions (red dashed arrow) involving the *Fgf8* AER enhancers now repositioned within the *Lbx1* TAD. **d** Whole-mount in situ hybridization for *Fgf8*, *Lbx1* and *Btrc* at E11.5. Expression was checked and confirmed in at least 3 or more homozygous embryos (at least *n* = 3 biological replicates). **e** Skeletal analysis of E18.5 limbs stained with alcian blue (cartilage) and alizarin red (bone). Hands and feet from wild-type, *Inv2, Inv1 ΔLbx1* and *Fgf8 AER enhancer KI* E18.5 embryos. At least *n* = 3 biological replicates.

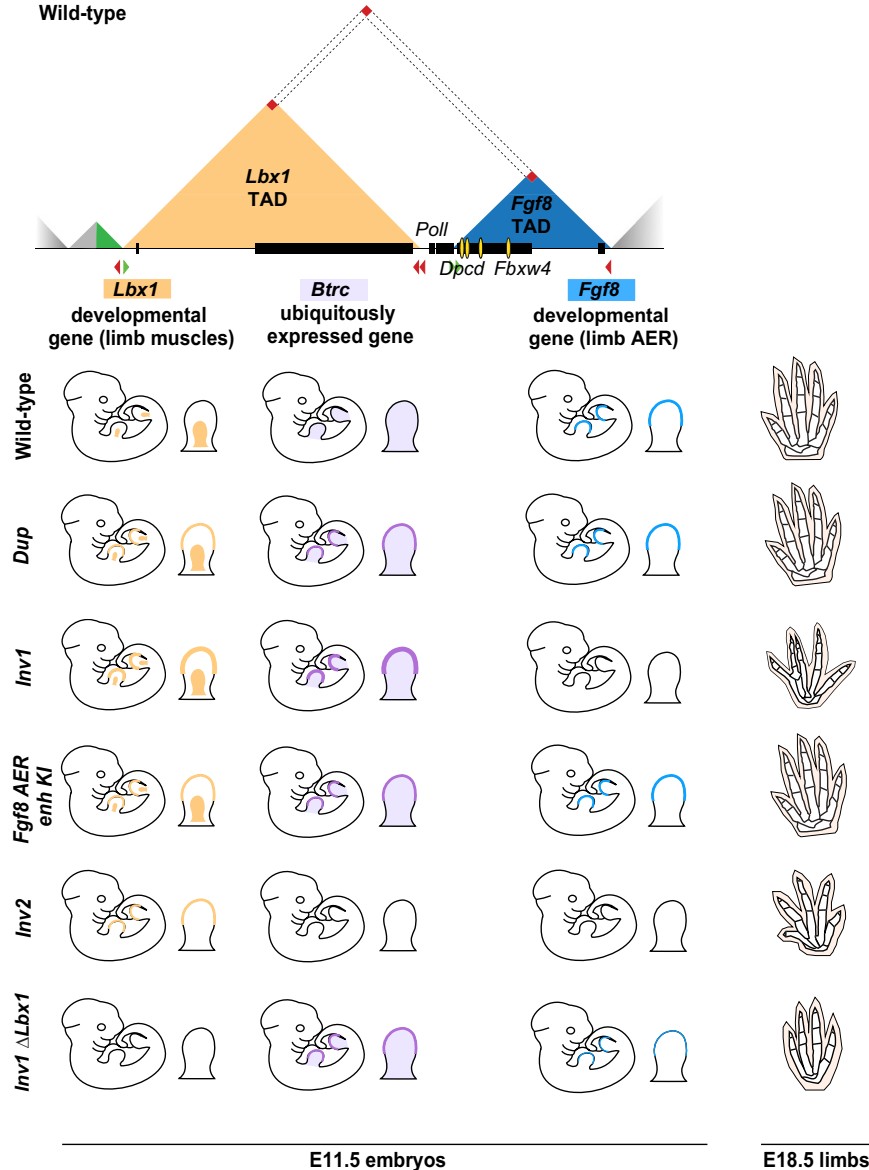

**Fig. 7 | Combinatorial effects on gene expression at the *Lbx1/Fgf8* locus leads to SHFM3 digit fusion phenotype.** Schematic overview of the *Lbx1-Fgf8* TADs (top) and *Lbx1*, *Btrc* and *Fgf8* expression in the developing limb at E11.5 and skeletal phenotype at E18.5 among the different structural rearrangements (bottom). Yellow and violet patterns represent endogenous expression in the limb bud and misexpression in the AER of *Lbx1* and *Btrc*, respectively. Turquoise pattern highlights *Fgf8* expression in the AER. Ectopic expression of both *Lbx1* and *Btrc* in the AER is observed for the *Dup, Inv1* and *Fgf8* AER enhancer KI (*Fgf8 AER enh KI*) mutant

embryos at E11.5. However, only the *Inv1* mutants, for which the level of *Lbx1* and *Btrc* expression in the AER is higher and *Fgf8* expression is missing, show a digit fusion phenotype at E18.5, reminiscent of human SHFM. Ectopic expression of either *Lbx1* alone (*Inv2* mutant) or *Btrc* alone (*Inv1 ΔLbx1* mutant) together with complete (*Inv2*) or partial (*Inv1 ΔLbx1*) loss of *Fgf8* AER expression is not enough to lead to the digit fusion SHFM-like phenotype but only results in an *Fgf8*-KO limb phenotype.

## Discussion

Split-hand/foot malformation (SHFM) is a congenital limb malformation affecting the central rays of the hands and/or feet (OMIM #183600, #313350, #600095, #605289, and #606708). The condition is a classic example of allelic heterogeneity, phenotypic variability, and pleiotropy[35].

Here, we focused on SHFM type 3 which has been associated with rearrangement at the *FGF8/LBX1* locus. cHi-C of the *Lbx1/Fgf8* locus in mouse limb bud showed that the region consists of two major TADs, each harbouring an important developmental gene with strikingly different expression patterns (Fig. 7). The *Fgf8* TAD contains a number of well-characterized enhancers that regulate *Fgf8* specific expression in the AER and in other tissues[28]. The neighbouring TAD contains *Lbx1*, a muscle-specific transcription factor expressed only in migrating muscle cells and *Btrc*, a ubiquitin ligase expressed at a low level in most limb cells and more stringently in the AER (Fig. 4c). The two regulatory TADs are separated by a boundary with divergent CTCF sites. A combination of cHi-C and 4C analysis demonstrated that the two domains are connected via a larger loop between the centromeric *Lbx1* TAD boundary and the telomeric *Fgf8* TAD boundary, resulting in a certain degree of contact of one domain with the other across the boundary (Fig. 1a). By re-engineering a human SHFM3-associated duplication in mice we observed ectopic interactions between the *Fgf8* AER enhancers and two other genes in the locus, *Lbx1* and *Btrc*. The same ectopic interactions were present in fibroblasts from a SHFM3 affected individual with duplication. In mice, this resulted in misexpression of *Lbx1* and *Btrc* in the AER in a typical *Fgf8* pattern. In contrast, *Fgf8* expression remained unchanged. Indeed, *Fgf8* was not included in the duplication and a fully intact *Fgf8* regulatory landscape remained, ensuring normal expression. The duplication resulted in the formation of a neo-TAD[4] which consisted of the *Fbwx4* gene and the *Fgf8* AER enhancers but no other genes. Our results suggest that the neo-TAD by itself does not have any consequences, as no ectopic interactions between *Fbxw4* and the *Fgf8* AER enhancers and/or misexpression of *Fbxw4* in the AER were detected (Fig. 2 and Supplementary Fig. 6b).

In a screen for structural variations in a cohort of patients with limb malformations, we also identified an inversion at the SHFM3 locus. Re-engineering of the human inversion in mice (*Inv1*) showed that the inversion resulted in a shift of regulatory elements from the *Fgf8* TAD into the *Lbx1* TAD[5] thereby giving rise to the equivalent regulatory effect as the *Dup*. Interestingly, as for the duplication, the inversion involved the four essential *Fgf8* AER enhancers (CE58, CE59, CE61, CE66) that are located in the introns of the *Fbxw4* gene (Fig. 1a, b). Previous experiments demonstrated that a loss of *Fgf8* expression in the AER upon deletion resulted in similar defects as conditional ablation of *Fgf8* in the limb[27,28]. In *Inv1* we also observed a loss of *Fgf8* expression and the corresponding *Fgf8* loss of function phenotype. Thus, a repositioning of the boundary by the inversion was sufficient to disconnect *Fgf8* from its enhancers resulting in a loss of *Fgf8* expression similar to the 4 AER enhancers deletion (Supplementary Fig. 7). The effect of *Inv1* was thus similar to the reported inversion at the *Sox9/Kcnj2* locus which resulted in a loss of *Sox9* expression due to a swap of *Sox9* regulatory elements from the *Sox9* TAD to the *Kcnj2* TAD[36].

In both SHFM-associated rearrangements, only *Lbx1* and *Btrc* were misexpressed in the AER. *Poll*, *Dpcd* and *Fbxw4* showed very low levels of expression in these cells. This strongly suggests that these genes were not involved in the pathogenesis of SHFM3. Our conclusion that *Lbx1* and *Btrc* were ectopically activated by the *Fgf8* AER enhancers was further supported by the insertion of the AER enhancers in the *Lbx1* TAD (*Fgf8 AER enh KI*). As in the duplication and in the inversion, the *Fgf8* AER enhancers were able to activate both *Btrc* and *Lbx1*, with the latter being more affected. In *Inv1* the interactions were possible through the repositioning of the *Fgf8* AER enhancers within the *Lbx1* TAD similar to the *Fgf8 AER enh KI* mutant. In the *Dup*, the situation is more complex. Here, the duplicated *Fgf8* AER enhancers are located within the neo-TAD but without their natural target promoter (*Fgf8*). This situation likely results in increased contacts across the leaky *Lbx1/Fgf8* boundary on the centromeric side as well as the newly positioned boundary telomeric of the neo-TAD, as indicated by the dashed line in Fig. 2d.

Thus, the common mechanism between the two SVs seems to be the ectopic expression of *Btrc* and *Lbx1* in the AER driven by *Fgf8* AER enhancers. To quantify the expression levels specifically in the AER, we performed single-cell RNA-sequencing of E11.5 limb buds. While the misexpression of *Lbx1* was similar in both mutants, we detected a higher AER misexpression of *Btrc* in the *Inv1* mutant compared to the *Dup* (Fig. 4c). Interestingly, two studies have reported SHFM3 malformation in patients with smaller duplication containing only BTRC alone[37] or BTRC and POLL[38], suggesting that abnormal expression of *Btrc* could be the main factor for the pathology of the human SHFM3 phenotype. Our data suggest that in mice, a high level of misexpression of both *Btrc* and *Lbx1* is required to generate the phenotype (Fig. 6). Finally, we cannot rule out the impact of genetic modifiers in different genetic backgrounds which might influence the manifestation of the phenotype, as observed in various congenital malformations such as for the role of TBX6 in scoliosis[39].

However, the precise pathomechanism through which *Btrc* and *Lbx1* cause the phenotype is still unclear. The Dactylaplasia (*Dac*) mouse mutant displays a similar phenotype to the one observed in some SHFM3 patients and the two corresponding alleles (*Dac1j* and *Dac2j*) map to the region with the duplications and inversion in SHFM3[40]. Based on these similarities, the mutant has long been thought as a model for human SHFM3, but *Dac1j* and *Dac2j* were shown to be associated with insertions of a MusD retrotransposon upstream of the *Fbxw4* gene and within intron 5 of *Fbxw4*, respectively[40]. The absence of any structural variations in these mutants and the differences between the nature of the human and mouse genomic abnormalities argue against a common pathogenesis. However, both the mouse and the human mutations involve the regulatory elements located within and around the *Fbxw4* gene possibly providing a common pathogenetic mechanism.

Similar to the reported misexpression of *Pax3* in an *Epha4*-like pattern in the limb in brachydactyly[3], misexpression of the muscle-specific transcription factor *Lbx1* and the ubiquitin ligase *Btrc* in the AER are likely to have effects on gene expression and thus the functionality of the AER. This interpretation is supported by our finding that *Lbx1* misexpression is associated with the activation of muscle genes in the AER cells (Fig. 5c). This effect seems to be stronger in the *Dup*, but given the low number of AER cells available for the differential expression analysis, we need to take this aspect with caution and consider the fact that other changes in gene expression could be stronger in *Inv1* due to the lack of *Fgf8* expression in this mutant. *Btrc* is part of a E3 ubiquitin-protein ligase complex which mediates the ubiquitination and subsequent proteasomal degradation of target proteins thereby participating in Wnt and BMP signalling[41], both important signalling components of the AER. The ectopic expression of this ubiquitin ligase may thus contribute to cellular dysfunction in the AER. The AER is the major signalling center for proximodistal growth and distal limb development. It is induced and maintained through the reciprocal interactions between the ectoderm and the underlying mesenchyme involving *Wnt-Bmp-Fgf* signaling pathways[42]. Disruption or malfunction of the AER results in diminished growth and thus hypoplasia/aplasia of digits, correlating with the SHFM phenotype[43]. Our results support the concept that SHFM is caused by AER defects, and in SHFM3 by misexpression of *Lbx1* and *Btrc* in the AER. Finally, we cannot exclude a potential contribution of a loss of *Fgf8* expression to the *Inv1* mutant phenotype.

Mice are commonly used as a model organism to study human disease because of the similarities in genetics, physiology and organ

development. However, inter-species differences exist and account for genotype-to-phenotype divergences between mouse and human[44]. When comparing CTCF binding sites in mouse and human at the *FGF8* locus, we noticed that the centromeric *Lbx1* TAD boundary is missing one binding site in reverse orientation in human in comparison to mouse (red arrow in Fig. 3a). This additional binding site in mouse could thus interfere with the levels of ectopic interactions between the *Fgf8* AER enhancers and *Lbx1* and *Btrc* in the *Dup* mutant, consequently reducing the level of misexpression in the AER. Indeed, misexpression levels related to the rearrangements may explain the absence of a striking phenotype in the *Dup* in comparison to *Inv1* which is associated with highest levels of *Lbx1* and *Btrc* misexpression, causing a mouse phenotype similar to the human phenotype.

Overall, this study offers insights into: i) the molecular phenotype of SHFM3-associated duplications and inversion, *ii)* gene regulation at the *Lbx1/Fgf8* locus in the context of 3D genome architecture and, iii) the consequences deriving from perturbations of the local chromatin structure. Notably, we provide a complex scenario by which SVs can cause disease. Indeed, we report in this study an example of SVs causing disease through a combined position effect mechanism resulting in ectopic gene misexpression involving multiple genes and a dose-dependent effect. To the best of our knowledge, the SVs-associated pathogenic mechanism reported in this study has not been shown previously.

In the era of whole genome sequencing (WGS) and the increased number of detected SVs, the medical interpretation of SVs and the prediction of their phenotypic consequences remain unsatisfactory. Thus, this study provides a conceptual framework when interpreting the pathogenic potential of these variant types.

## Methods
The experiments were not randomized. The investigators were not blinded to allocation during experiments and outcome assessment.

### Ethical approvals
All animal procedures were conducted as approved by the local authorities (LAGeSo Berlin) under license numbers G0243/18 and G0176/19. All animal experiments followed all relevant guidelines and regulations.

The study design and conduct complied with all relevant regulations regarding the use of human participants and was conducted in accordance with the criteria set by the Declaration of Helsinki. Congenital limb malformations were observed at clinical genetic centres in Germany and in Italy. Given the nature of their phenotype, patients were offered additional diagnostic investigations in accordance with regulations for studies on human subjects at the respective centres. Informed consent to participate to the study and to publish clinical data was obtained from all patients (or their legal guardian). The study protocol was approved by the ethics committee at the Charité Universitätsmedizin Berlin, Germany.

### Mouse embryonic stem cell targeting
Culture and genome editing of mouse embryonic stem cells (mESCs) was performed as described previously[24,36]. Briefly, G4 (129sv/C57Bl6 hybrid) mESCs were seeded on a monolayer of CD1 feeder cells and transfection was performed via FuGENE technology (Promega, Cat. #E5911) using 8ug of each CRISPR construct of interest delivered through the pX459 vector (which contains also a Puromycin-resistance cassette). After 24 h, cells were split onto DR4 Puromycin-resistant feeders and selection with Puromycin was carried on for 2 days. Resistant growing clones were then picked and grown into 96-well-plates on CD1 feeders. All clones were genotyped by PCR (Supplementary Table 2). Positive clones were expanded, their genotyping further confirmed by PCR, qPCR (Supplementary Table 3) and Sanger sequencing of PCR products, and used for further experiments only if

the successful modification could be verified. qPCR results were analyzed using the QuantStudio 7 Flex Real-Time PCR Software (version 1.7.1, Applied Biosystems).

The size and position of the human structural variations (hg19/GRCh37) were converted to the mouse genome (mm9/NCBI37) using the UCSC liftOver tool (https://genome.ucsc.edu/cgi-bin/hgliftOver). A list of single guide RNAs (sgRNAs, designed using http://crispr.mit.edu/ platform) used for CRISPR/Cas9 genome editing is given in Supplementary Table 1. For the *Fgf8-AER-enh-KI* mutant, a 6993 bp DNA fragment containing the 4 *Fgf8*-AER enhancers with a deletion of the CTCF site in enhancer CE58 was synthetised and subsequently cloned into a pUC vector by *Genewiz*. Homology arms targeting the *Lbx1* locus were then cloned using Gibson assembly and the resulting vector was transfected in mESCs together with the pX459-sgRNA for CRISPR/Cas9 homology-directed-repair (HDR).

### Generation of mice
Mice were generated from genome-edited mESCs by diploid or tetra-ploid aggregation[45] and genotyped by PCR and qPCR analysis. Cells were seeded on CD1 feeders and grown for 2 days before aggregation. Female mice of the CD1 strain were used as foster mothers. Several founder animals for each mouse line were used for establishing line stock with variable intercrosses between single founder and C57BL/6 wild-type animals. Selection of animals for analysis and breeding was random. For all animal experiments, no choice of sample size was applied. The data collection was performed according to the stage of each sample and investigators were not blinded since mouse breeding and analysis required knowledge about the genotype at hand. Routine bedding, food and water changes were performed. Mice were housed in a centrally controlled environment with a 12 h light: 12 h dark cycle, temperature of 20–22.2 C°, and humidity of 30–50%.

### Whole mount in situ hybridization (WISH)
RNA expression in E11.5 mouse embryos from wild-type and mutants was assessed by WISH using digoxigenin (DIG)-labelled antisense riboprobes for *Lbx1*, *Btrc*, *Poll*, *Dpcd*, *Fbxw4* and *Fgf8* transcribed from linearized gene-specific probes (PCR DIG Probe Synthesis Kit; Roche, Cat. #11636090910). Primers for probe generation are listed in Supplementary Table 5. Embryos were collected and fixed overnight in 4% PFA in PBS, then washed twice for 30 min in PBS with 0.1% Tween (PBST), dehydrated for 30 min each in 25%, 50% and 75% methanol in PBST and stored at −20 °C in 100% methanol. For WISH, genotyped embryos were rehydrated on ice in reverse methanol/PBST steps, washed twice in PBST, bleached in 6% $H_2O_2$ in PBST for 1 h and washed in PBST. Embryos were washed in PBST and refixed for 20 min with 4% PFA in PBS, 0.2% glutaraldehyde and 0.1% Tween. After further washing steps with PBST, embryos were incubated at 68 °C in L1 buffer (50% deionized formamide, 5× SSC, 1% SDS, 0.1% Tween 20 in diethyl pyrocarbonate (DEPC), pH 4.5) for 10 min. For prehybridization, embryos were incubated for 2 h at 68 °C in hybridization buffer 1 (L1 with 0.1% tRNA and 0.05% heparin), while for subsequent probe hybridization, embryos were incubated overnight at 68 °C in hybridization buffer 2 (hybridization buffer 1 with 0.1% tRNA, 0.05% heparin and 1:500 DIG probe). The next day, unbound probes were washed away through repeated washing steps: 3 × 30 min at 68 °C with L1, L2 (50% deionized formamide, 2× SSC, pH 4.5, 0.1% Tween 20 in DEPC, pH 4.5) and L3 (2× SSC, pH 4.5, 0.1% Tween 20 in DEPC, pH 4.5). For signal detection, embryos were treated for 1 h with RNase solution (0.1 M NaCl, 0.01 M Tris, pH 7.5, 0.2% Tween 20, 100 μg ml$^{-1}$ RNase A in $H_2O$), followed by washing in TBST 1 (140 mM NaCl, 2.7 mM KCl, 25 mM Tris–HCl, 1% Tween 20, pH 7.5) and blocking for 2 h at room temperature in blocking solution (TBST 1 with 2% calf serum and 0.2% bovine serum albumin). Overnight incubation with Anti-Dig antibody conjugated to alkaline phosphatase (1:5,000) at 4 °C (Roche, Cat.#11093274910) was followed by 8 × 30 min washing steps at room temperature with TBST

 

2 (TBST with 0.1% Tween 20 and 0.05% levamisole–tetramisole) and left overnight at 4 °C. Embryos were finally stained after equilibration in AP buffer (0.02 M NaCl, 0.05 M MgCl₂, 0.1% Tween 20, 0.1 M Tris–HCl and 0.05% levamisole–tetramisole in $H_2O$) 3 × 20 min, followed by staining with BM Purple AP Substrate (Roche, Cat. #11442074001). The stained embryos were imaged using a Zeiss SteREO Discovery V12 microscope and Leica DFC420 digital camera.

## Skeletal preparation
E18.5 fetuses were kept in H2O for 1–2 h at room temperature and heat shocked at 65 °C for 1 min. The skin was gently removed, together with the abdominal and thoracic viscera using forceps. The fetuses were then fixed in 100% ethanol overnight. Afterwards, alcian blue staining solution (150 mg l − 1 alcian blue 8GX in 80% ethanol and 20% acetic acid) was used to stain the cartilage overnight. After 24 h fetuses were rinsed and postfixed in 100% ethanol overnight, followed by 24 h incubation in 0.2% KOH in H2O for initial clearing. The next day fetuses were incubated in alizarin red (50 mg l − 1 alizarin red S in 0.2% potassium hydroxide) to stain the bones overnight. Following this, rinsing and clearing was done for several days using 0.2% KOH. The stained embryos were dissected in 25% glycerol, imaged using a Zeiss SteREO Discovery V12 microscope and Leica DFC420 digital camera, and subsequently stored in 80% glycerol.

## Capture Hi-C
E11.5 mouse limb buds were prepared in 1x PBS and dissociated by trypsin treatment for 10 min at 37 °C, pipetting every 2 min to obtain a single-cell suspension. Trypsin was stopped by adding 5x volume of 10% FCS/PBS. The solution was then filtered through 40 µm cell strainer to remove cell debris, cells were centrifuged at 130 x g for 5 min and the pellet was then resuspended in 5 ml 10% FCS/ PBS. 5 ml of freshly prepared 4% formaldehyde in 10% FCS/PBS (final concentration 2%) were added to perform crosslinking and samples were incubated in rotation for 10 min at room temperature. The crosslinking reaction was stopped by adding 1 ml 1.425 M glycine on ice, followed by centrifugation at 240 x g and 4 °C for 8 min, pellet resuspension in 5 ml freshly prepared, cold lysis buffer (final concentration of 10 mM Tris pH 7.5, 10 mM NaCl, 5 mM MgCl2, 0.1 M EDTA, and 1 × cOmplete™ protease inhibitors (Roche, Cat. #04693132001) and incubation for at least 10 min on ice. Nuclei were pelleted by centrifugation at 425 x g at 4 °C for 5 min, washed with 1x PBS, aliquoted in 1.5 ml tubes with 2.5–5×10⁶ nuclei each, followed by supernatant removal, snap-freezing and storage at −80 °C.

Snap-frozen pellet was resuspended in 360 µl of water, mixed with 60 µl of 10x DpnII buffer (Thermo Fisher Scientific, Cat. #ER1701) and incubated for 5 min at 37 °C in a thermomixer at 900 rpm. 15 µl of 10% SDS was then added and the samples were incubated for 1 h at 37 °C and 900 rpm, with occasional pipetting to help dissolving the nuclei aggregates. Next, 150 µl of 10% Triton X-100 were added, followed by 1 h of incubation at 37 °C and 900 rpm. After adding 600 µl of 1x DpnII buffer, a 10 µl aliquot was taken as undigested control and stored at 4 °C and 400 units of restriction enzyme DpnII were added to the samples. After 4 h of digestion additional 200 units of DpnII enzyme were added and the samples were incubated overnight at 37 °C with shaking at 900 rpm. After overnight incubation, a 10 µl aliquot was taken as digested control and stored at 4 °C, while the samples were supplemented with further 200 units of DpnII enzyme for additional four hours. Meanwhile, the undigested and digested controls were tested. For this, each 10 µl aliquot was mixed with 85 µl 10 mM Tris pH 7.5 and 2 µl RNase A (10 mg/ml) and incubated for 1 h at 37 °C. Chromatin was decrosslinked by adding 5 µl proteinase K (10 mg/ ml) and incubation at 65 °C for 4 h. The DNA was then extracted by adding 100 µl phenol-chloroform. Samples were mixed by inverting the tubes and centrifuged for 10 min at 18500 x g at room temperature. The upper water phase was transferred into new 1.5 ml tubes and analysed

on a 1% agarose gel. After correct validation, restriction enzyme in the original samples still under digestion was heat-inactivated at 65 °C for 20 min. The samples were transferred to 50 ml Falcon tubes and 700 µl 10x ligation buffer (Thermo Fisher Scientific, Cat. #EL0021) were added. The volume was filled up to 7 ml with water and 50 units of T4 DNA ligase (Thermo Fisher Scientific, Cat. #EL0021) were added. The ligation mix was incubated overnight at 4 °C in rotation. The next day, a 100 µl aliquot of religated DNA was collected and analysed on an agarose gel as described above. Upon successful ligation, samples were de-crosslinked by adding 30 µl of proteinase K (10 mg/ml) and incubating overnight at 65 °C. Then, 30 µl RNase A (10 mg/ml) were added and the samples were incubated for 45 min at 37 °C. The DNA was then extracted by adding 7 ml phenol-chloroform. The solution was mixed by inverting the tube and the water phase was separated by centrifugation at 1500 x g for 15 min at room temperature. DNA was precipitated by adding the following reagents to the water phase: 7 ml water, 1.5 ml 2 M NaAc pH 5.6, 140 µg glycogen, 35 ml 100 % ethanol. The solution was mixed and placed at −80 °C overnight or until the samples were completely frozen. The sample was then thawed and centrifuged for 20 min at 8350 g and 4 °C. DNA pellet was washed with 30 ml cold 70% ethanol and centrifuged 15 min at 3300 g at 4 °C. Dried pellet was dissolved in 150 µl 10 mM Tris pH 7.5 at 37 °C. The final 3 C library was checked on 1% agarose gel and subsequently used for capture Hi-C preparation. 3 C libraries were sheared using a Covaris sonicator (duty cycle: 10%; intensity: 5; cycles per burst: 200; time: 6 cycles of 60 s each; set mode: frequency sweeping; temperature: 4–7 °C). Adaptors were added to the sheared DNA and amplified according to Agilent instructions for Illumina sequencing. The library was hybridized to the custom-designed SureSelect beads and indexed for sequencing following Agilent instructions. SureSelect enrichment probes were designed over the genomic interval chr19:44,440,000-46,400,000 (mm9) using the online tool of Agilent: SureDesign (https://earray.chem.agilent.com/suredesign/). Probes were covering the entire genomic region and were not designed specifically in proximity of DpnII sites. All samples (3 biological replicates per experiment plus one technical, except for *Inv1* het where no technical replicate was produced, and for *Dup* het where only one biological replicate was used) were then sequenced with the Hiseq4000 or Novaseq6000 Illumina technology according to the standard protocols and with around 400 million 50 bp, 75 bp (Hiseq4000) or 100 bp (Novaseq6000) paired-end reads per sample.

## Capture Hi-C data processing
In a pre-processing step, the reads in fastq files were trimmed to 50 bp, if necessary, to obtain the same initial read length for all samples. Afterwards, mapping, filtering and deduplication of paired-end sequencing data was performed using the HiCUP pipeline v0.8.1[46] (no size selection, Nofill: 1, Format: Sanger). The pipeline was set up with Bowtie2 v2.4.2[47] for mapping short reads to reference genome mm9. For merging biological replicates, the final bam files produced by the HiCUP pipeline were joined. Juicer tools v1.19.02[48] d was used to generate binned and KR normalized[49,50] contact maps from valid and deduplicated read pairs. For the generation of cHi-C maps, only read-pairs referring to the region of interest (chr19:44,440,001-46,400,000) and with MAPQ ≥ 30 were considered. We used cHi-C maps with 5 kb bin size. For the sample with a duplication it is noted, that the matrix balancing of the KR normalization affects the signal in the duplicated region, because it scales down the signal intensities in this region to fit to the other regions of the cHi-C map. In order to consider the duplicated region explicitly, we created for the *Dup* mutant sample an additional version of the cHi-C map by applying LOIC normalization[51] (python package iced v.0.5.10[52]) to raw count map, which has the aim to retain the effects of the increased copy number. For the LOIC normalization the copy number was set to 4 for all bins overlapping the

duplication, and to 2 for the other bins. For five matrix rows/columns with low coverage, the count values were removed prior to LOIC-normalization.

Subtraction maps were computed from KR-normalized cHi-C maps, which were normalized in a pairwise manner before subtraction as follows. To account for differences between two maps in their distance-dependent signal decay, the maps were scaled jointly across their sub-diagonals. Therefore, the values of each sub-diagonal of one map were divided by the sum of this sub-diagonal and multiplied by the average of these sums from both maps. Afterwards, the maps were scaled by $10^6$/total sum. For the computation of scaling factors, the duplicated as well as the inverted regions were excluded in both maps. cHi-C maps were visualized as heatmaps with linear scale, with very high values being truncated to improve visualization.

## Virtual 4C

Virtual 4C-like interaction profiles were generated for individual viewpoints from the same bam files also used for the cHi-C maps. Paired-end reads with MAPQ ≥ 30 were considered in a profile, when one read mapped to the defined viewpoint region and the other one outside of it. Contacts of a viewpoint region were counted per restriction fragment. The count profile was binned to a 1 kb grid. In case a fragment overlapped more than one bin, the counts were distributed proportionally. Afterwards, the binned profile was smoothed by averaging within a sliding window of 5 kb size and scaled by $10^3$/sum of counts within the enriched region. The viewpoint region itself and a margin ±5 kb around it were excluded from the computation of the scaling factor. Interaction profiles were generated with custom Java code using v2.12.0 (https://samtools.github.io/htsjdk/).

## RNA expression analysis

E11.5 limb buds were microdissected from wild-type and mutant embryos (at least $n$ = 3) and immediately frozen in liquid nitrogen. Total RNA was isolated using the RNeasy Mini Kit (Qiagen, Cat. #74004). For RNA-sequencing, samples were poly-A enriched and sequenced with the Novaseq6000 Illumina technology according to the standard protocols and with around 100 million 100 bp paired-end reads per sample. For RT-qPCR, cDNA was generated using 1 ug of extracted RNA and reverse transcribed using Superscript III (Life Technologies, Cat. # 18080093) with oligo(dT)$_{25}$ as template primers. RT-qPCR was performed using the Biozym Blue S'Green qPCR Mix Separate ROX (Biozym, Cat. # 331416 S). Relative expression levels were normalized to the geometric mean of the Ct for housekeeping genes *Rrm2* and *RplpO*, with the ΔΔCt method and data were shown as normalized to WT values. Primers used are listed in Supplementary Table 4.

## RNA-seq data processing

Reads were mapped to the mouse reference genome (mm9) using the STAR mapper[53] (splice junctions based on RefSeq; options: –alignIntronMin 20–alignIntronMax 500000–outFilterMultimapNmax 5–outFilterMismatchNmax 10–outFilterMismatchNoverLmax 0.1). Reads per gene were counted as described previously[3], and used for differential expression analysis with the DEseq2 package[54].

## scRNA-seq

scRNA-seq experiments were performed in single replicates that were jointly processed to avoid batch effects. E11.5 limb buds of wild-type, *Dup* and *Inv1* embryos were microdissected in 1xPBS at room temperature. A single-cell suspension was obtained by incubating the tissue for 10 min at 37 °C in 200 µl Gibco trypsin-EDTA 0.05% (Thermo Fisher Scientific, Cat. #25300054) supplemented with 20 µl 5% BSA. Trypsinization was then stopped by adding 400 µl of 5% BSA. Cells were then resuspended by pipetting, filtered using a 0.40 µm cell, washed once with 0.04% BSA, centrifuged for 5 min at 150 x *g*, then

resuspended in 0.04% BSA. The cell concentration was determined using an automated cell counter (Bio-Rad) and cells were subjected to scRNA-seq (10x Genomics, Chromium Single Cell 3' v2; one reaction per time point and per strain has been used) aiming for a target cell recovery of up to 10,000 sequenced cells per sequencing library (time point and strain). Single-cell libraries were generated according to the 10x Genomics instructions with the following conditions: 8 PCR cycles were run during cDNA amplification and 12 PCR cycles were run during library generation and sample indexing to increase library complexity. Libraries were sequenced with a minimum of 230 million 75 bp paired-end reads according to standard protocols.

## scRNA-seq data processing

The Cell Ranger pipeline v.3 (10x Genomics Inc.) was used for each scRNA-seq sample in order to de-multiplex the raw base call files, to generate the fastq files, and to perform the alignment against a custom mouse reference genome mm9 to create the unique molecular identifier (UMI) count matrix. Only cells with more than 2000 detected genes and less than 10% of mitochondrial UMI counts were considered for further analysis. In addition, Scrublet[55] was used to identify potential doublets in our dataset. Cells with a Scrublet score higher than 0.2 were filtered out. Each sample was normalized independently using the SCT method[56] implemented in Seurat[57] R package and then, these were integrated using the Seurat3 IntegrateData CCA-based (canonical correlation analysis) approach considering the top 2,000 most variable genes. The first 20 principal components of this joint dataset were calculated and used for UMAP projection and reconstruction of the cell-cell similarity graph. To delimitate the major limb bud cell types, we used the Louvain algorithm implemented in the Seurat3 function FindClusters and the expression of well-known marker genes for the limb-comprising cell types. For the AER cluster, we used the marker genes *Fgf8* and *En1*. Differential gene expression between the wild-type (Bl6) vs *Dup* and wild-type (G4) vs *Inv1* hindlimb cells was estimated by modelling the gene expression as a function of the genotype across cells. This analysis was done from a new embedding of the samples separately: on one hand wild-type (Bl6) and *Dup*, on the other hand wild-type (G4) and *Inv1*. We fitted a quasi-Poisson distribution to calculate the effect of the *Dup* or the *Inv1* genotype on the gene expression distribution of each gene using the monocle3 strategy[58]. We tested that such effect was not equal to 0. We use this condition association effect to rank the genes and identified the pathways associated with the top 200 genes using the Enrichr tool from the Maayan lab[59]. Finally, we confirmed that the top perturbed genes associated to the muscle pathways are more frequently expressed in the *Dup* AER cells.

## 4C-seq

For 4C-seq libraries, $5 \times 10^6$–$10^7$ cells were used. The fixation and lysis were performed as described in the Capture Hi-C section. After the first digestion with DpnII, sticky ends were religated in a 50 ml falcon tube (700 µl 10X ligation buffer, 7 ml H2O, 50 U T4 DNA ligase; overnight at 16 °C) and DNA de-cross linked and cleaned as described in the Capture Hi-C section. Next, a second digestion (150 µl sample, 50 µl 10× Csp6I buffer, 60 U Csp6I (Thermo, Cat. # FD0214) 295 µl H2O; overnight at 37 °C) and another re-ligation were performed. For all viewpoints, DNA was purified using a PCR clean up Kit (Qiagen, Cat. #28104) and 1.6 µg DNA was amplified by PCR (*LBX1* Viewpoint: read-primer 5'-TCTATATGCTACCATGATC-3', secondary-primer 5'-GATGAACTGGAATACCCA-3'; *FGF8* Viewpoint: read-primer 5'-AGGGTGCGTTCCAAGATC-3', secondary-primer 5'-GGTGGCCTGGATGGAAGT-3'; *BTRC* Viewpoint: read-primer 5'-CAACGCAGCGCCCGGATC-3', secondary-primer 5'-CTGGGAATGAGGACCTAGGGC-3'). For the library reaction, primers were modified with TruSeq adapters: Adapter1 5'-CTACACGACGCTCTTCCGATCT-3' and Adapter2 5'-CAGAC GTGTGCTCTTCCGATCT-3'. Between 50 and 200 ng were used as input of a single 4C PCR reaction

depending on the complexity. The reaction was performed in a 50 µl volume using the Expand Long Template System (Roche, Cat. # 11681834001) and 29 reaction cycles. After the PCR all reactions were combined and the DNA purified with a PCR clean up Kit. All samples were then sequenced with the Novaseq6000 Illumina technology according to the standard protocols and with around 5 million 100 bp single-end reads per sample.

### 4C-seq data processing
Sequencing reads were mapped, normalized and smoothed with pipe4C[60] using the reference genome GRCh37 and default settings. All viewpoints were performed in replicates and as quality measure >70% of reads were mapped within a size range of 1 Mb and >80% within 100 kb around the viewpoint.

### Human material
Skin biopsies were collected from SHFM3 patients and controls by standard procedures. Fibroblasts were cultured in DMEM supplemented with 10% fetal calf serum (Gibco, Cat. #A4766801), 1% L-glutamine (Lonza, Cat. #BE17-650E) and 1% penicillin/streptomycin (Fisher, Cat. #BP295950).

### Reporting summary
Further information on research design is available in the Nature Portfolio Reporting Summary linked to this article.

## Data availability
Sequencing data generated in this study have been deposited in the Gene Expression Omnibus (GEO) database and are available under accession code GSE197404.

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

## Acknowledgements

This study was supported by grants from the Deutsche For- schungsgemeinschaft (MU 880/16-1, MU 880/20-1) to S.M. We thank the transgenic unit, sequencing core and animal facility of Max Planck Insti- tute for Molecular Genetics for technical assistance, Ute Fischer for technical support and Norbert Brieske for help with whole mount in situ hybridizations and image processing. We would like to thank all members of the Mundlos laboratory and Tugce Aktas for continuous support and discussion. We would like to particularly thank Dario Lupiáñez, Chiara Anania, Lila Allou and Jane Skok for critical reading of the manuscript.

## Author contributions

G.C., S.M. and M.S. conceived the project. G.C and J.G. performed most experiments with technical support of M.Fr., R.Fa. and M.Fe. L.W. per- formed morula aggregation. R.S. performed cHi-C computational ana- lysis and generated virtual 4C maps. C.A.P-M. performed Single Cell RNA-seq computational analysis. S.A. performed and analysed 4C-seq. B.T. oversaw high-throughput sequencing. M.S., E.P., R.Fi., F.N., M.G., and O.Z. examined the patients, interpreted the clinical data and pro- vided the clinical information together with biological samples. G.C., J.G. and S.M. wrote the manuscript with input from all the authors.

## Funding

## Competing interests

The authors declare no competing interests.
