## [Peer Review File · Nature Communications]

Combinatorial effects on gene expression at the Lbx1/Fgf8 locus resolve Split-Hand/Foot Malformation type 3REVIEWER COMMENTS

Reviewer #1 (Remarks to the Author):

[Ed: Please see attached document from Reviewer 1. Contact me if you have issues accessing the file.]

The proposed wet lab experiments are "nice to have" and are not necessary for this reviewer. However, they would shore up more strongly the role of the individual enhancers and strengthen assertions of dosage effects by the authors.

In this exciting manuscript from the Mundlos & Spielman groups they reprise much of their ground breaking work in showing how noncoding components of TAD structure are highly influential on genetic diseases.

They identify a key locus implicated in Split-Hand/Foot Malformation type 3, (SHFM3) specifically the Lbx1/Fgf8 locus where the posit that multiple duplications and inversions play a significant role in the etiology of the disease. Importantly they recreate in mouse models, not only duplication and inversion, but insertion of discrete regulatory elements into the locus to assign agency for the phenotype to the specific noncoding components of the TAD structure. The authors perform several experiments to solidify this assertion of a role of the enhancer elements in SHFM3 and conclude that it is not only misexpression but also dosage that affects the developing limb.

The manuscript is tightly constructed, and streamlined with good evidence, in general, to exclude alternative conclusions. This works well to explain misexpression. Dosage effects are more complex to tease out and perhaps outside the scope of this study. For this reason, it may mean toning down the certainty of dosage in some parts of the current manuscript. Overall this manuscript is highly suitable for publication in this journal with some relatively minor revisions and comments to be addressed.

1. Early in the introduction the authors state “SVs [*structural variants*] can alter the copy number of regulatory elements or modify the 3D genome by disrupting higher-order chromatin organization. This can give rise to ectopic enhancer-promoter contacts, gene misregulation and ultimately disease⁵. However, the general applicability of this concept remains under investigation.” Though the one reference is provided here, a review, it would be helpful to narrow in on primary work tightly linking pathogenicity to for example; “higher order chromatin rearrangements leading to unambiguous pathogenic dosage effects” if such example(s) exist in the literature.
2. In the same paragraph as (1) the authors state the following: “*Studies in Drosophila, for example, demonstrated that the disruption of TADs and enhancer-promoter interactions are not necessarily accompanied by changes in expression, indicating a high degree of robustness of the genome against such events.*” More emphasis has to be placed on the fact that this reference provided and the robustness referred to is in the context of the drosophila genome (so far). Perhaps editing thus is sufficient: “*indicating a high degree of robustness of the drosophila genome.....*”
3. In figure 1a the authors refer to two chromatin regulatory domains in two black dashed lines. However there are three black dashed circles in Figure 1a. Could the authors more explicitly clarify what chromatin regulatory domains they are referring to as being in the cartoon. Next to Fig1a. I recommend enlarging the cartoon to be at least the size of the Hi-C map. In addition Fig 1b and the written descriptions provided of the published duplications could benefit substantially from a cartoon next to figure 1b similar to 1a. Specifically a “zoom in” view in the cartoon of the especially interesting boundary region

between Lbx1 and Fgf8 and the Fgf8 AER enhancers where the common patho-mechanism is thought to originate would be highly useful.

4. Since the genes *Poll* and *Dpcd* are located in the boundary region separating Lbx1 and Fgf8, is it explicitly clear whether the expression of these genes dissolves the boundaries on one or both sides.
5. In Figure 2 the authors describe the neo-TAD formation as follows: *The neo-TAD, indicated by the blue/green triangle between the two Lbx1 TADs (yellow), contained only Fbxw4, the Fgf8 AER enhancers and a small (~38 kb) region flanking the centromeric side the Lbx1 TAD boundary, thus unlikely to have a pathological effect by itself.* The directionality of the contacts sites emanating from the Fbxw4 to the duplicated region seem to “pierce” the CTCF boundary and sufficient to contact Lbx1 and Btrc. Furthermore the Fgf8 enhancers 58 to 73 seem to remain in this “blue” area where Fbxw4 is located and are able to engage in these long range contacts. How certain are the authors of this “portion” of the TAD to be non-pathogenic alone? No experiment is requested, however if the “blue” portion of the duplicated region in Figure 2d was lost would there be no pathogenicity?
6. In Figure 3B where examples of ectopic interaction are shown comparing control and the duplication at LBX1 and BTRC, the peak heights shown by the red arrows are not so convincing to those unskilled in the art of 4C-seq. Is there a way to augment the evidence of enhancer driven long-range contacts by a loss of function of experiment in the human fibroblasts with the SHFM3 duplication? For example in the human fibroblast lines using an LNA/Gapmer or CRISPR deletion to one or more of the FGF8 AER enhancers. Understandably duplication complicates this experiment, however, most CRISPR deletions are inefficient. While the LNA/Gapmer may target all the enhancers the 4C-seq would show a decline in contacts at all regions the enhancers are implicated in targeting (including those with the unconvincing peaks indicated by red arrows). Ideally an inefficient CRISPR deletion to one copy of the duplicated enhancers would be ideal though albeit a low efficiency and more challenging experiment. Therefore an attempt at the easier LNA/Gapmer experiment would be encouraged by this reviewer.
7. Figure 4C contains some highly interesting data on the effects of duplication and cell type specific gene expression. What makes the *Dpcd* ectopic expression in the muscle and DV ectoderm occur only in the duplication? What effect of the duplication can adequately explain this?
8. While the authors do conduct a difficult enhancer insertion experiment using CRISPR in Figure 5, the enhancer logic of the individual enhancers in directing expression to specific tissues of the limb bud as revealed in Fig 4C is obscured by this experiment. Perhaps conducting the enhancer silencing (using LNA/Gapmers) experiment proposed

in point 6 in fibroblasts from these mice would be informative of the enhancer logic at play.

9. While the authors in their introduction and abstract emphasize dosage effects (through duplication) the combinatorial effects outlined in Figure 6 do not seem to bear this out. Rather inversions seem to be far more deleterious. Is this because the number of copies in the human disease is far higher (4 or 5) and the model in mice is only creating a duplication (2 copies)?
10. In supplemental figure 4a the authors note that they also noticed the embryos exhibited underdeveloped head structure most likely as a consequence of midbrain-hindbrain boundary enhancers involved into this larger inversion. Could there be more explicit dosage effects at play and to what extent does enhancer copy number or gene copy number (through duplications) play a role in the underdevelopment of the head structure?

Reviewer #2 (Remarks to the Author):

Excellent study that provides insights in the pathogenesis of a duplication and an inversion in the LBX1/FGF8 locus in a hand and foot malformation.

Questions/Critique

1. The engineered mice studied with dup and inv1 were homozygous for these structural variants. However, the phenotype of SHFM3 in humans is due to heterozygous structural variants. Did the authors look at heterozygous dup and inv1 mice? Is there a phenotype in the heterozygous inv1 mice? How do the TADs look in the heterozygous mice?
2. The statements of lines 173-178 are speculative; this could be better clarified.
3. The differences in fig 3a and 3b are not striking.
4. The schematic of fig 2d is theoretical, and this needs to be stated.
5. The dose-dependent part of the phenotype is not well explained; I suggest to develop this further.
6. Could the authors speculate why the dup mice (homozygous and heterozygous) do not have a phenotype?
7. Is the nature of the normal allele (eQTLs and expression of the relevant genes) in the human SHFM3 a contributing factor to the appearance or not of the phenotype?

Reviewer #3 (Remarks to the Author):

In the present study, Giulia Cova et al investigated the functional consequences of the chromosomal rearrangements associated with Split-Hand/Foot Malformation type 3 (SHFM3) on chromatin conformation and gene expression in vivo in transgenic mice. SHFM3 is a congenital limb defect that is characterized by tandem duplications at the LBX1/FGF8 locus, with obscure underlying pathogenesis.

The Authors show that the LBX1/FGF8 locus consists of two separate, but interacting, regulatory domains. They re-engineer a SHFM3-associated duplication and a newly reported inversion in mice, using CRISPR/Cas-mediated genome editing and report that these rearrangements result in restructuring of chromatin architecture. This leads to an ectopic activation of Lbx1 and Btrc gene expression in the apical ectodermal ridge (AER) in an Fgf8-like pattern. Artificial repositioning of the AER-specific enhancers of Fgf8 is sufficient to induce misexpression of Lbx1 and Btrc. The Authors conclude that the SHFM3 phenotype is the result of a combinatorial effect on gene misexpression and gene dosage in the developing limb.

General Comments:

The causal relationships between chromosomal rearrangements, alterations of chromatin architecture, and disturbances of gene regulation have been intensely studied and discussed in recent years. On one hand, Hi-C studies profiling the genome 3D chromatin structure have highlighted evolutionarily conserved

Topologically Associating Domains (TADs) that correlate with gene expression, and evidence from mouse models and human disease has directly linked TADs to gene regulation. On the other hand, a number of recent studies have questioned the impact of 3D chromatin domains on gene expression, emphasizing the need for a deeper integration of 3D chromatin structure with genomic and functional read-outs. Given

these intense debates within the scientific community devoted to the study of genome organization and gene regulation, the present manuscript is certainly timely and is also of high interest. The study is well written, the technology employed is cutting-edge, the figures are elegant and masterfully crafted, and, more importantly, the results reported contribute new insights into the molecular mechanisms that underpin SHFM3, a maiming limb malformation. Overall, the manuscript provides an interesting conceptual framework for how genomic rearrangements can cause gene misexpression, which, in turn, results in limb birth defects of obscure molecular pathogenesis.

That said, there are serious concerns regarding some of the results reported in this study, especially regarding the alleged differences in contacts reported in the 4C experiments in controls and mutants, as detailed below. There is also uneasiness about the conclusions that are reached regarding some experiments, which appear to be based on overinterpretation of some of the data. Lastly, and importantly, the overall logic of the study and the sequence of the reported findings are at times hard to follow and would benefit from re-organization.

Specific Points of Concern:

- This Reviewer believes that the overall sequence of the findings reported in the present manuscript would greatly benefit from re-organization. From the point of view of the experimental logic, the study would acquire a better “flow” if the Authors would report the characterization of the limb phenotypes in their control and mutant mouse embryos immediately after showing the engineering of the human SHFM3 duplication (Dup) and the inversion (Inv1) in mice, and before illustrating the chromatin conformation landscapes in control and mutants. Such re-organization of the data would require considerable swapping of figures and adjustments to the text, but it would improve the rationale and logical sequence of the described findings.

Indeed, showing first the characterization of the mouse phenotypes would provide a functional read-out of high significance to understand and support the biological relevance and impact of the observed ectopic interactions -or lack thereof- in mutants. These genome-wide changes would be placed within a developmental context, as they are presented to the reader.

It is notable that this same group adopted a similar strategy in a previous paper (where the limb phenotypes in control and genome-edited mutant mouse embryos were illustrated immediately after showing the engineering of the chromosomal rearrangements: Lupianez et al, Cell, 2015).

- Virtual 4C analysis of the Capture Hi-C (Chi-C) data sets with viewpoints on Lbx1 and Fgf8 in E11.5 mouse wild-type limb buds strongly emphasizes how the interactions of these genes are mainly restricted within their respective chromatin regulatory domains (interactions highlighted in yellow in Fig. 1a). Fgf8 and Lbx1 are located very close to the centromeric and telomeric boundaries, respectively, as illustrated in the virtual 4C graph, making it easy to appreciate the interactions with the TAD boundary region (highlighted in violet in Fig. 1a) that define the two domains. Despite being two separated domains, the Lbx1 and Fgf8 TADs are also connected via an additional loop (Fig. 1a, top dashed circle in cHi-C map). This interaction across the boundary is very clearly visible in the virtual 4C analysis (highlighted in pink in Fig. 1a) indicating that subpopulations of limb bud cells can have different configurations, and also that there is a certain degree of leakiness of the Lbx1/Fgf8 boundary, leading to a certain level of intermingling of the two domains in spite of the very different expression patterns of Lbx1 and Fgf8. The data shown in Fig. 1 are clear and convincing. In summary, the Authors demonstrate that the Virtual 4C analysis constitutes a strong addition to their analysis of the Chi-C data sets and adds important and detailed information on all occurring interactions. In light of the substantially added resolution obtained by implementing the Virtual 4C analyses, as shown in Fig. 1, it is not clear why the Authors did not employ the same rigorous strategy when analyzing the ectopic interactions that occur upon structural rearrangements at the Lbx1/Fgf8 locus in E11.5 mouse limb buds with either Duplication or Inversion (shown in Fig. 2). Implementing Virtual 4C analyses also in the mutant limbs, to be included in Fig. 2, would help substantially to convincingly demonstrate “new” or “increased” interactions or “loss” of interactions in mutant limbs. In the opinion of this reviewer, it would be important to show the Virtual 4C analyses in mutant limb buds within Fig. 2.

- The Authors conduct 4C-seq in human fibroblasts and state that there is the presence of ectopic interactions involving LBX1 and BTRC in patients with the Duplication (Fig. 3). The difference between the 4C maps in the fibroblasts derived from patients versus controls appears to be very minimal (regions highlighted by red arrows within the yellow rectangles in Fig. 3b). Lacking a formal statistical analysis, which is not routinely performed in comparisons of different 4C-seq results, it is very hard to

judge whether these alleged differences in contacts are significant. In the absence of any possible statistical analysis, the Authors should employ an orthogonal method to prove the presence of meaningful differences in contacts between patients and controls. For example, 3C-seq (i.e. one versus one) in control and patient fibroblasts could be used in at least two independent replicates in order to provide robust and convincing data.

- The Authors state that comparison of the human data sets (discussed above) with mouse virtual 4C plots generated from cHi-C maps of Duplication (Dup) confirmed that the same ectopic interactions identified in human cells existed also in their Dup mutant mice (Fig. 3c). Regrettably this Reviewer has the same problem with Fig. 3c as discussed above in Fig. 3b, i.e. the alleged differences are minimal at best, making it hard to attribute them real significance. Especially in consideration of the finding that Inv1 -but not Dup- rearrangements result in a SHFM-like limb phenotype in mouse embryos, it is particularly awkward to understand how the alleged ectopic interactions in Dup mutants could have a functional impact, in the absence of any phenotype.
- The Authors show that both Dup and Inv1 chromatin rearrangements have an impact on gene expression by bulk RNA-seq in E11.5 limb buds (Fig. 4a). However, they conduct single cell RNAseq only in Dup limb buds (Fig. 4b,c). Given that later within the "Results" section of their manuscript the Authors demonstrate that Inv1 -but not Dup- rearrangements result in a SHFM-like phenotype, this Reviewer finds it awkward to understand why the single cell RNA-seq was done only in Dup limb buds. It would have been important to perform single cell RNA-seq also in Inv1 limb buds to check gene expression levels changes at the single cell level in mice with a clear SHFM-like phenotype. The Authors should at least explain why that was not done and should give a rationale for this choice.
- Representations of whole-mount in situ hybridizations for Fgf8, Lbx1 and Btrc on E11.5 limb buds (in controls, Dup and Inv1) should be increased in size, especially related to the limb bud insets (Fig. 5a). The current panels make it very difficult to evaluate the expression levels of the respective genes and their perturbations. Also, higher contrast and sharpness of the in situ hybridization signal would improve the quality of these panels.

Minor Issues

- Supplementary Fig. 4 contains very important results showing that misexpression of both Lbx1 and Btrc is required to develop a SHFM phenotype. Therefore, this Figure should be included as a main Figure in the text. Given that authors only have 6 main Figures in the text, Suppl Fig. 4 should be moved into the main text.
- Line 262-264: The Authors should further discuss the relevance of the myogenic genes being abnormally activated in the epithelial cells of Dup and Inv1 mutant mice.

Point-by-point response to the reviewers:

We sincerely thank the reviewers for their evaluation of our work. Considering their helpful and constructive comments, we have now performed additional experiments, which we think will address their concerns. We provide here a point-by-point response to the reviewers' questions/comments. Below, are the original comments in blue and our responses in black. Relevant changes, additions and important aspects have been highlighted in yellow in the revised manuscript.

Reviewer #1 (Remarks to the Author):

In this exciting manuscript from the Mundlos & Spielman groups they reprise much of their ground breaking work in showing how noncoding components of TAD structure are highly influential on genetic diseases. They identify a key locus implicated in Split-Hand/Foot Malformation type 3, (SHFM3) specifically the *Lbx1/Fgf8* locus where the posit that multiple duplications and inversions play a significant role in the etiology of the disease. Importantly they recreate in mouse models, not only duplication and inversion, but insertion of discrete regulatory elements into the locus to assign agency for the phenotype to the specific noncoding components of the TAD structure. The authors perform several experiments to solidify this assertion of a role of the enhancer elements in SHFM3 and conclude that it is not only misexpression but also dosage that affects the developing limb. The manuscript is tightly constructed, and streamlined with good evidence, in general, to exclude alternative conclusions. This works well to explain misexpression. Dosage effects are more complex to tease out and perhaps outside the scope of this study. For this reason, it may mean toning down the certainty of dosage in some parts of the current manuscript. Overall, this manuscript is highly suitable for publication in this journal with some relatively minor revisions and comments to be addressed.

1. Early in the introduction the authors state “SVs [structural variants] can alter the copy number of regulatory elements or modify the 3D genome by disrupting higher-order chromatin organization. This can give rise to ectopic enhancer-promoter contacts, gene misregulation and ultimately disease⁵. However, the general applicability of this concept remains under investigation.” Though the one reference is provided here, a review, it would be helpful to narrow in on primary work tightly linking pathogenicity to for example; “higher order chromatin rearrangements leading to unambiguous pathogenic dosage effects” if such example(s) exist in the literature.

Thank you for the comment. We have taken this into account and added two more references to the manuscript that specifically refer to this mechanism (Line 61).

2. In the same paragraph as (1) the authors state the following: “Studies in *Drosophila*, for example, demonstrated that the disruption of TADs and enhancer-promoter interactions are not necessarily accompanied by changes in expression, indicating a high degree of robustness of the genome against such events.” More emphasis has to be placed on the fact

that this reference provided and the robustness referred to is in the context of the drosophila genome (so far). Perhaps editing thus is sufficient: “indicating a high degree of robustness of the drosophila genome.....”

We agree and we have clarified this in the manuscript. We have added a recent study from our lab (Schöpflin et al. *Nat. Commun.* 2022) that investigates congenital chromothrypsis cases and comes to a similar conclusion (Line 63-65).

3. In figure 1a the authors refer to two chromatin regulatory domains in two black dashed lines. However, there are three black dashed circles in Figure 1a. Could the authors more explicitly clarify what chromatin regulatory domains they are referring to as being in the cartoon.

We have provided now a better explanation in the manuscript (Lines 118-122), together with a better highlight of the bigger domain with black double dashed lines in Fig. 1A as we used in the cartoon in Fig. 2B.

Next to Fig1a. I recommend enlarging the cartoon to be at least the size of the Hi-C map. In addition Fig 1b and the written descriptions provided of the published duplications could benefit substantially from a cartoon next to figure 1b similar to 1a. Specifically, a “zoom in” view in the cartoon of the especially interesting boundary region between *Lbx1* and *Fgf8* and the *Fgf8* AER enhancers where the common patho-mechanism is thought to originate would be highly useful.

To the best of our understanding, the reviewer is here referring to Fig. 2 and not Fig.1, as no cartoons were present in Fig. 1A. As suggested by the reviewer, we have now reorganized this Figure and provided a “zoom in” to highlight the boundary region of interest between *Lbx1* and *Fgf8* TADs. We hope this makes it clearer.

4. Since the genes *Poll* and *Dpcd* are located in the boundary region separating *Lbx1* and *Fgf8*, is it explicitly clear whether the expression of these genes dissolves the boundaries on one or both sides.

Thanks for pointing this out. It is not known whether expression of these genes contributes to the boundary. We detected their expression in our RNA-seq E11.5 developing limb data, therefore a potential contribution cannot be completely excluded. We have now added this aspect in the manuscript (Line 124-126).

5. In Figure 2 the authors describe the neo-TAD formation as follows: “The neo-TAD, indicated by the blue/green triangle between the two *Lbx1* TADs (yellow), contained only *Fbxw4*, the *Fgf8* AER enhancers and a small (~38 kb) region flanking the centromeric side the *Lbx1* TAD boundary, thus unlikely to have a pathological effect by itself”.

The directionality of the contacts sites emanating from the *Fbxw4* to the duplicated region seem to “pierce” the CTCF boundary and sufficient to contact *Lbx1* and *Btrc*. Furthermore, the *Fgf8* enhancers 58 to 73 seem to remain in this “blue” area where *Fbxw4* is located and are able to engage in these long range contacts. How certain are the authors of this “portion” of the TAD to be non-pathogenic alone? No experiment is requested, however if the “blue” portion of the duplicated region in Figure 2d was lost would there be no pathogenicity?

The neo-TAD is now indicated as a “blue-orange” triangle and in the manuscript we have now adapted the part which had a confusing terminology. In the Result section of the manuscript,

we wrote now: “Interestingly, we observed a stripe of increased interactions (indicated by black arrowheads) involving the part of the *Fgf8* TAD containing the *Fgf8* AER enhancers comprised in the duplication (blue bar in Fig. 2C) and a region immediately upstream of the *Lbx1* centromeric TAD boundary (orange bar in Fig. 2C). This reflected the formation of a neo-TAD, as illustrated in the schematic of the predicted linear configuration (Fig 2D).” (lines 179-184). In the discussion, we now wrote (lines 360-364): “The duplication resulted in the formation of a neo-TAD which consisted of the *Fbxw4* gene and the *Fgf8* AER enhancers but no other genes. Our results suggest that the neo-TAD by itself does not have any consequences, as no ectopic interactions between *Fbxw4* and the *Fgf8* AER enhancers and/or misexpression of *Fbxw4* in the AER were detected (Fig. 2 and Supplementary Fig. 6B).” We believe the interactions within the neo-TAD itself (in this case between the *Fgf8* enhancers and *Fbxw4*) do not have any impact. Indeed, we did not see any misexpression of *Fbxw4* in the AER. However, the neo-TAD provides enhancers for establishing ectopic interactions outside (as indicated by red dotted arrows on the new Fig. 2D). This likely results in the ectopic activation of *Lbx1* and *Btrc*. Furthermore, as mentioned above, we modified Fig. 2 and detailed the schematic to hopefully facilitate the understanding. The other “blue” portion of the duplicated region (Figure 2) shouldn’t have any pathological effect and we think is not responsible for the ectopic expression of *Btrc* and *Lbx1* since the structure is similar as in wild-type.

6. In Figure 3B where examples of ectopic interaction are shown comparing control and the duplication at *LBX1* and *BTRC*, the peak heights shown by the red arrows are not so convincing to those unskilled in the art of 4C-seq. Is there a way to augment the evidence of enhancer driven long-range contacts by a loss of function of experiment in the human fibroblasts with the SHFM3 duplication? For example in the human fibroblast lines using an LNA/Gapmer or CRISPR deletion to one or more of the *FGF8* AER enhancers. Understandably duplication complicates this experiment, however, most CRISPR deletions are inefficient. While the LNA/Gapmer may target all the enhancers the 4C-seq would show a decline in contacts at all regions the enhancers are implicated in targeting (including those with the unconvincing peaks indicated by red arrows). Ideally an inefficient CRISPR deletion to one copy of the duplicated enhancers would be ideal though albeit a low efficiency and more challenging experiment. Therefore an attempt at the easier LNA/Gapmer experiment would be encouraged by this reviewer.

Thank you for the comment and suggestion. We did not perform the proposed AER loss of function experiment in human fibroblast. We believe fibroblast is not the best model to show a change in terms of gene expression as the genes of interest are not or very lowly expressed in this tissue. The mouse limb bud seemed to be a better model for this purpose. However, as mentioned by the reviewer, a loss of function experiment in the duplicated background would be elegant but is extremely challenging as it is impossible to target specifically the enhancers in the duplicated region. Although we could not perform the exact experiment suggested by this reviewer, we have nevertheless generated more data to hopefully clarify this point.

- First, we have now improved the visualization of our 4C-seq data in human fibroblast to make the ectopic contact clearer (see below and Fig. 3):

- We have also performed comparison to control regions to confirm that the peak heights observed at the *Fgf8* enhancers region in the duplication are significantly higher than those observed in the control samples (see below and Supplementary Fig. 4):

- The mouse virtual 4C are now shown in the same way improving the visualization of the ectopic contact in *Dup* and *Inv1* mutants (see below and Figure 3 and Supplementary Fig. 4):

- We have deleted in a wild-type background the 4 previously characterized AER enhancers (Marinic et al. 2013) using CRISPR in mouse embryonic stem cells and generated embryos for further analysis. The results of this experiment are detailed below in point 8.

7. Figure 4C contains some highly interesting data on the effects of duplication and cell type specific gene expression. What makes the *Dpcd* ectopic expression in the muscle and DV ectoderm occur only in the duplication? What effect of the duplication can adequately explain this?

The *Dpcd* gene is included in the duplicated region. In the homozygous *Dup* mutant, there are thus 4 copies of the *Dpcd* gene which are expressed, although at a rather low level in the developing limb at E11.5. This explains its double expression level seen in the *Dup* mutant by bulk RNA-sequencing (Fig. 4A) and by single-cell RNA-sequencing in the different cell types (AER, DV ectoderm, mesenchyme and muscle, Fig. 4C).

8. While the authors do conduct a difficult enhancer insertion experiment using CRISPR in Figure 5, the enhancer logic of the individual enhancers in directing expression to specific tissues of the limb bud as revealed in Fig 4C is obscured by this experiment. Perhaps conducting the enhancer silencing (using LNA/Gapmers) experiment proposed in point 6 in fibroblasts from these mice would be informative of the enhancer logic at play.

We performed loss of function of the same 4 AER enhancers (58, 59, 61 and 66) we used for the knock-in in Fig. 5 using CRISPR in mouse embryonic stem cells. E11.5 mouse embryos were generated from this and we conducted expression analysis experiments. As shown in Supplementary Fig. 7 and below, this deletion abolishes the expression of *Fgf8* in the developing limb in the same way it was abolished in the *Inv1* mutant. This shows that these enhancers are the major regulator of *Fgf8* expression in the AER. In contrast, the increase of *Lbx1* and *Btrc* expression is specific to the *Inv1* mutant.

Also, as mentioned above, on Fig. 3 we showed the quantification of the virtual 4C integrated signal from a 125kb region comprising the *Fgf8* AER enhancers. We demonstrate a gain of interaction with *Lbx1/Btrc* which is specific to this region (Figure 3 and Supplementary Fig. 4).

9. While the authors in their introduction and abstract emphasize dosage effects (through duplication) the combinatorial effects outlined in Figure 6 do not seem to bear this out. Rather inversions seem to be far more deleterious. Is this because the number of copies in the human disease is far higher (4 or 5) and the model in mice is only creating a duplication (2 copies)?

We describe two distinct observations: increase in gene expression due to gene dosage, due to the duplication, and ectopic expression, due to repositioning of the enhancers in both, the *Dup* and the *Inv1*. Since the ectopic expression is higher in the *Inv1* than the *Dup*, we used the term “dosage effects” for the ectopic expression, however, we acknowledge that this may be confusing. We apologize for not making that clear initially and we have now clarified this point throughout the manuscript.

- Gene dosage: in the *Dup* mouse mutant, there is a change in the dosage of several genes at the locus (*Lbx1*, *Btrc*, *Poll*, *Dpcc4* and *Fbxw4*) (Fig. 4A). Since the duplication is homozygous in our mice, the copy number of these 5 genes increases from 2 to 4. However, this doesn't

seem to affect the development of the limb. This gene dosage effect doesn't exist in the *Inv1* mutant since this is a balanced rearrangement.

- Ectopic expression: we observed ectopic expression, i.e. in regions where the genes are normally not expressed, of *Lbx1* and *Btrc* in the AER in both *Dup* and *Inv1* mutants. We argue that this ectopic expression is caused by the ectopic contact of the 4 *Fgf8* AER enhancers with *Lbx1* and *Btrc* promoters (Fig. 2 and 3). Interestingly, it seems that these ectopic expressions of *Lbx1* and *Btrc* are not similar between the two mutants. By bulk RNA-sequencing (Fig. 4A) we detected a gain of expression of *Btrc* and *Lbx1* in the limb bud of *Dup* and *Inv1* mutants. However, these data were based on bulk sequencing and are thus not representative for tissue specific expression. To overcome this problem, we generated single-cell RNA-sequencing data for the *Dup* (first version of the manuscript) and then also for the *Inv1* mutant (added in the new version of the manuscript) (Fig. 4C). These data show the *Lbx1* expression in the AER is similar in the *Dup* and the *Inv1* mutants. In contrast, the *Btrc* expression in the AER is much higher in the *Inv1* mutant than in the *Dup*. We made similar observation using in situ hybridization (Fig. 5), leading to the conclusion that the inversion resulted in a stronger activation. We hypothesize that this might explain the discrepancy in phenotype between the two mutants (Lines 391-399). However, we cannot rule out the fact that the decrease of *Fgf8* expression might also have a role in the development of the digit phenotype.

10. In supplemental figure 4a the authors note that they also noticed the embryos exhibited underdeveloped head structure most likely as a consequence of midbrain-hindbrain boundary enhancers involved into this larger inversion. Could there be more explicit dosage effects at play and to what extent does enhancer copy number or gene copy number (through duplications) play a role in the underdevelopment of the head structure?

We observed underdeveloped head structure only for the embryos harbouring the larger inversion (*Inv2*, Figure 6D). *Fgf8* brain enhancers have been identified in the region closer to *Fgf8* which is part of the *Inv2* but not *Inv1* (Hörnblad et al 2021). We thus suggested that in this larger inversion, the brain enhancers of *Fgf8* have been repositioned far away from *Fgf8* and thus leads to a decrease of *Fgf8* expression in the brain and the underdeveloped phenotype. Although we did not perform any expression analysis or cHi-C data in the brain as this is not the scope of this paper, this is a reasonable explanation as we made a similar observation in the limb with the *Inv1*: the AER enhancers of *Fgf8* have been re-positioned and *Fgf8* expression is thus abolished in the AER resulting in underdeveloped forelimbs and hindlimbs. The *Dup* includes two of the characterized *Fgf8* brain enhancers (Hörnblad et al 2021), suggesting *Btrc* and *Lbx1* might also be ectopically expressed in this tissue. However, we did not detect this using whole mount in situ hybridization where we detect only ectopic expression in the AER. Importantly, we never observed any underdeveloped head structure in the *Dup* mutant such as the one observed for the *Inv2*, suggesting this phenotype is a result of loss of *Fgf8* expression rather than any increased copy number of gene or enhancers.

Reviewer #2 (Remarks to the Author):

Excellent study that provides insights in the pathogenesis of a duplication and an inversion in the LBX1/FGF8 locus in a hand and foot malformation.

Questions/Critique

1. The engineered mice studied with dup and inv1 were homozygous for these structural variants. However, the phenotype of SHFM3 in humans is due to heterozygous structural variants. Did the authors look at heterozygous dup and inv1 mice? Is there a phenotype in the heterozygous inv1 mice? How the TADs look in the heterozygous mice?

We have now added the heterozygous data for both *Dup* and *Inv1* mutants: the E18.5 skeletal preparations (Fig. 1D and Supplementary Fig. 3), the E11.5 WISH (Fig. 5A and Supplementary Fig. 6A) and the cHi-C maps (Supplementary Fig. 4). In summary, no skeletal malformations were observed in heterozygous mice for both *Dup* and *Inv1*, but we could detect misexpression of *Lbx1* and *Btrc* in the AER already, as well as ectopic interactions in the cHi-C maps involving the *Fgf8* AER enhancers and the *Lbx1* TAD, though with a lower intensity and frequency. This could explain the absence of phenotype as a certain threshold of misexpression (as discussed also in lines 438-440) needs to be reached.

2. The statements of lines 173-178 are speculative; this could be better clarified.

We clarified this in lines 185-193, graphically in Fig. 2D and this is also discussed in lines 385-390.

3. The differences in fig 3a and 3b are not striking.

We are now providing a better visualization of both our 4C-seq data in human fibroblast and in the virtual 4C data from cHi-C in mouse in order to make the ectopic contact more evident (See Fig. 3 and answer to the similar comment 6. from reviewer 1). Additionally, we performed quantification of the integrated 4C signal over a 125kb region (Fig. 3). We thus showed that the region containing the *Fgf8* enhancers in the duplication (human and mouse) and the inversion (mouse) exhibit a gain of interaction with both *Lbx1* and *Btrc* promoter which is not present in two control regions outside of the *Fgf8* TAD (Fig. 3 and Supplementary Fig. 4).

4. the schematic of fig 2d is theoretical, and this needs to be stated.

We clarified this and now state that the schematics are a prediction in lines 183-184.

5. The dose-dependent part of the phenotype is not well explained; I suggest to develop this further.

As discussed in lines 391-399, 438-440, we think that a certain threshold of misexpression of *Lbx1* and *Btrc* in the AER needs to be reached to develop the phenotype. Indeed, *Inv1* hom is the only mutant showing a SHFM-like phenotype in correlation with higher level of misexpression compared to the heterozygous and the *Dup* mutants. Additionally, our data (Fig. 6 and Fig. 7) show that on top of the misexpression dosage levels, both *Lbx1* and *Btrc* are required to be ectopically expressed to develop the phenotype.

More details to this are also provided in the answer to the following point 6 and we have developed this further with our new single-cell sequencing data in the manuscript.

6. Could the authors speculate why the dup mice (homozygous and heterozygous) do not have a phenotype ?

For both mutants, we observed ectopic expression of *Lbx1* and *Btrc* in the AER. However, the in-situ hybridization (Fig. 5A) and newly generated single-cell RNA-seq (Fig. 4C) data showed a stronger ectopic expression in the *Inv1* mutant than in the *Dup* mutant.

We observed that *Btrc* ectopic expression in the AER was much stronger in the *Inv1* than in the *Dup* mutant (see below). This is interesting to put in perspective with a new study (Qiu et al. 2022) reporting cases of SHFM3 in a family with a micro-duplication affecting only *BTRC*.

Since we observed an SHFM-like phenotype only in the *Inv1*, we propose that a certain threshold of ectopic expression in the AER needs to be reached to cause the phenotype. This is particularly true when comparing *Inv1* homozygous and heterozygous mutants. Also, when looking at our knock-in of the AER enhancers next to *Lbx1*, *Btrc* is not as ectopically expressed as in the *Inv1* mutant. Furthermore, as discussed also in lines 430-440, we have to consider that, despite the high level of similarities between the mouse and human genomes, some inter-species differences exist. Because of this, the level of misexpression required to develop the phenotype in mouse might be higher than in humans where the SHFM3 phenotype manifests already in heterozygosity, though still with a certain clinical variability ranging from a very mild to more severe phenotype. Finally, we of course can't rule out that the decrease of *Fgf8* expression plays a role in the *Inv1* phenotype.

7. Is the nature of the normal allele (eQTLs and expression of the relevant genes) in the human SHFM3 a contributing factor to the appearance or not of the phenotype?

This is an interesting thought that has – to our knowledge – not been addressed so far.

Reviewer #3 (Remarks to the Author):

In the present study, Giulia Cova et al investigated the functional consequences of the chromosomal rearrangements associated with Split-Hand/Foot Malformation type 3 (SHFM3) on chromatin conformation and gene expression in vivo in transgenic mice. SHFM3 is a congenital limb defect that is characterized by tandem duplications at the LBX1/FGF8 locus, with obscure underlying pathogenesis.

The Authors show that the LBX1/FGF8 locus consists of two separate, but interacting, regulatory domains. They re-engineer a SHFM3-associated duplication and a newly reported inversion in mice, using CRISPR/Cas-mediated genome editing and report that these rearrangements result in restructuring of chromatin architecture. This leads to an ectopic activation of *Lbx1* and *Btrc* gene expression in the apical ectodermal ridge (AER) in an *Fgf8*-like pattern. Artificial repositioning of the AER-specific enhancers of *Fgf8* is sufficient to induce misexpression of *Lbx1* and *Btrc*. The Authors conclude that the SHFM3 phenotype is the result of a combinatorial effect on gene misexpression and gene dosage in the developing limb.

General Comments:

The causal relationships between chromosomal rearrangements, alterations of chromatin architecture and disturbances of gene regulation have been intensely studied and discussed in recent years. On one hand, Hi-C studies profiling the genome 3D chromatin structure have highlighted evolutionarily conserved

Topologically Associating Domains (TADs) that correlate with gene expression, and evidence from mouse models and human disease has directly linked TADs to gene regulation. On the other hand, a number of recent studies have questioned the impact of 3D chromatin domains on gene expression, emphasizing the need for a deeper integration of 3D chromatin structure with genomic and functional read-outs. Given these intense debates within the scientific community devoted to the study of genome organization and gene regulation, the present manuscript is certainly timely and is also of high interest. The study is well written, the technology employed is cutting-edge, the figures are elegant and masterfully crafted, and, more importantly, the results reported contribute new insights into the molecular mechanisms that underpin SHFM3, a maiming limb malformation. Overall, the manuscript provides an interesting conceptual framework for how genomic rearrangements can cause gene misexpression, which, in turn, results in limb birth defects of obscure molecular pathogenesis.

That said, there are serious concerns regarding some of the results reported in this study, especially regarding the alleged differences in contacts reported in the 4C experiments in controls and mutants, as detailed below. There is also uneasiness about the conclusions that are reached regarding some experiments, which appear to be based on overinterpretation of some of the data. Lastly, and importantly, the overall logic of the study and the sequence of the reported findings are at times hard to follow and would benefit from re-organization.

Specific Points of Concern:

- This Reviewer believes that the overall sequence of the findings reported in the present manuscript would greatly benefit from re-organization. From the point of view of the experimental logic, the study would acquire a better “flow” if the Authors would report the characterization of the limb phenotypes in their control and mutant mouse embryos immediately after showing the engineering of the human SHFM3 duplication (Dup) and the inversion (Inv1) in mice, and before illustrating the chromatin conformation landscapes in control and mutants. Such re-organization of the data would require considerable swapping

of figures and adjustments to the text, but it would improve the rationale and logical sequence of the described findings.

Indeed, showing first the characterization of the mouse phenotypes would provide a functional read-out of high significance to understand and support the biological relevance and impact of the observed ectopic interactions -or lack thereof- in mutants. These genome-wide changes would be placed within a developmental context, as they are presented to the reader.

It is notable that this same group adopted a similar strategy in a previous paper (where the limb phenotypes in control and genome-edited mutant mouse embryos were illustrated immediately after showing the engineering of the chromosomal rearrangements: Lupianez et al, Cell, 2015).

We thank the reviewer for this comment. We re-organized the manuscript according to the suggestion and agree that this leads to a better narrative of our study. The skeletal prep showing the phenotype now appear in Fig. 1D and Supplementary Fig.3 (including the heterozygotes as suggested by reviewer 2). We then describe the observed changes in the 3D conformation (Fig. 2 and 3). Last, we link those to changes of expression (Fig. 4 and 5). We have also adapted this in the text.

- Virtual 4C analysis of the Capture Hi-C (Chi-C) data sets with viewpoints on *Lbx1* and *Fgf8* in E11.5 mouse wild-type limb buds strongly emphasizes how the interactions of these genes are mainly restricted within their respective chromatin regulatory domains (interactions highlighted in yellow in Fig. 1a). *Fgf8* and *Lbx1* are located very close to the centromeric and telomeric boundaries, respectively, as illustrated in the virtual 4C graph, making it easy to appreciate the interactions with the TAD boundary region (highlighted in violet in Fig. 1a) that define the two domains. Despite being two separated domains, the *Lbx1* and *Fgf8* TADs are also connected via an additional loop (Fig. 1a, top dashed circle in cHi-C map). This interaction across the boundary is very clearly visible in the virtual 4C analysis (highlighted in pink in Fig. 1a) indicating that subpopulations of limb bud cells can have different configurations, and also that there is a certain degree of leakiness of the *Lbx1*/*Fgf8* boundary, leading to a certain level of intermingling of the two domains in spite of the very different expression patterns of *Lbx1* and *Fgf8*. The data shown in Fig.1 are clear and convincing. In summary, the Authors demonstrate that the Virtual 4C analysis constitutes a strong addition to their analysis of the Chi-C data sets and adds important and detailed information on all occurring interactions. In light of the substantially added resolution obtained by implementing the Virtual 4C analyses, as shown in Fig. 1, it is not clear why the Authors did not employ the same rigorous strategy when analyzing the ectopic interactions that occur upon structural rearrangements at the *Lbx1*/*Fgf8* locus in E11.5 mouse limb buds with either Duplication or Inversion (shown in Fig. 2). Implementing Virtual 4C analyses also in the mutant limbs, to be included in Fig. 2, would help substantially to convincingly demonstrate “new” or “increased” interactions or “loss” of interactions in mutant limbs. In the opinion of this reviewer, it would be important to show the Virtual 4C analyses in mutant limb buds within Fig. 2.

In the new version of Fig. 3, we now provide the virtual 4C also for the mutant *Inv1*, together with a better overview and detailed virtual 4C analysis for both *Dup* and *Inv1* (see below, more details in the next comments). However, we decided to not add these data to Fig. 2 as it contained already a lot of information. We show the data in Fig. 3, allowing a better comparison also with the 4C-seq data for our Dup clinical case (fibroblasts from the patient carrying the SHFM inversion were not available).

- The Authors conduct 4C-seq in human fibroblasts and state that there is the presence of ectopic interactions involving LBX1 and BTRC in patients with the Duplication (Fig. 3). The difference between the 4C maps in the fibroblasts derived from patients versus controls appears to be very minimal (regions highlighted by red arrows within the yellow rectangles in Fig. 3b). Lacking a formal statistical analysis, which is not routinely performed in comparisons of different 4C-seq results, it is very hard to judge whether these alleged differences in contacts are significant. In the absence of any possible statistical analysis, the Authors should employ an orthogonal method to prove the presence of meaningful differences in contacts between patients and controls. For example, 3C-seq (i.e. one versus one) in control and patient fibroblasts could be used in at least two independent replicates in order to provide robust and convincing data.

We agree with the reviewer and we have now improved the visualization of our 4C-seq data in human fibroblast to make the ectopic contact clearer. We did not perform 3C-seq but rather decided to quantify the integrated 4C signal in the region of interest (*Fgf8* enhancers) and compare it to the signal in two control regions of the same size. As shown in the comment 6. from reviewer 1 and in Figure 3A and Supplementary Fig. 4A, the integrated signal in the region of interest (125kb) shows a gain of interaction with the *Lbx1* and *Btrc* promoters whereas we do not see any changes for the two control regions (125kb each). We hope that this new representation and quantification of our data will convince to reviewer that this effect, although subtle, is real.

- The Authors state that comparison of the human data sets (discussed above) with mouse virtual 4C plots generated from cHi-C maps of Duplication (Dup) confirmed that the same ectopic interactions identified in human cells existed also in their Dup mutant mice (Fig. 3c). Regrettably this Reviewer has the same problem with Fig. 3c as discussed above in Fig. 3b, i.e. the alleged differences are minimal at best, making it hard to attribute them real significance. Especially in consideration of the finding that *Inv1* -but not Dup- rearrangements result in a SHFM-like limb phenotype in mouse embryos, it is particularly awkward to understand how the alleged ectopic interactions in Dup mutants could have a functional impact, in the absence of any phenotype.

We now present the mouse virtual 4C data in the same way as the human 4C-seq data and performed a quantification of the integrated signal which was also applied to two control regions. This shows that there is a similar increase in contact between the *Lbx1/Btrc* promoters and the *Fgf8* enhancers in the *Dup* and *Inv1* mutants. However, as discussed in the manuscript, two features of the virtual 4C data are present in the *Inv1* but not in the *Dup* mutants: 1) an ectopic contact between the *Fgf8* promoter and the *Lbx1/Btrc* promoter (see black arrow on the Figure 3D and figure below) and 2) a loss of contacts between the *Fgf8* promoter and its enhancer region (Figure 3D and figure below).

To further investigate the difference in the phenotype of the *Inv1* mutants vs. the *Dup* mutants, we examined the impact of these SVs on gene expression (Figure 4 and 5). Using these data, we can link the observations from the virtual 4C with expression changes: 1) the ectopic expression of *Btrc* in the AER is stronger in the *Inv1* than in the *Dup* and 2) *Fgf8* expression is lost in the AER of the *Inv1* but not in the *Dup* mutants. These differences can explain why the phenotype appears only in the *Inv1* and not in the *Dup*. See also discussion lines 391-399.

- The Authors show that both Dup and Inv1 chromatin rearrangements have an impact on gene expression by bulk RNA-seq in E11.5 limb buds (Fig. 4a). However, they conduct single cell RNAseq only in Dup limb buds (Fig. 4b,c). Given that later within the “Results” section of their manuscript the Authors demonstrate that Inv1 -but not Dup- rearrangements result in a SHFM-like phenotype, this Reviewer finds it awkward to understand why the single cell RNA-seq was done only in Dup limb buds. It would have been important to perform single cell RNA-seq also in Inv1 limb buds to check gene expression levels changes at the single cell level in mice with a clear SHFM-like phenotype. The Authors should at least explain why that was not done and should give a rationale for this choice.

We agree with the reviewer and now provide single cell data also for the *Inv1* mutant (Fig. 4 B-C, Fig.5 and Supplementary Fig.5). Interestingly, these new data show that the ectopic expression of *Btrc* is stronger in the *Inv1* mutant than in the *Dup* (Fig. 4C and see below). This confirms the observation we made from the in situ hybridization data (Fig.5A) although those are not as quantitative as sequencing.

- Representations of whole-mount in situ hybridizations for *Fgf8*, *Lbx1* and *Btrc* on E11.5 limb buds (in controls, Dup and Inv1) should be increased in size, especially related to the limb bud insets (Fig. 5a). The current panels make it very difficult to evaluate the expression levels of

the respective genes and their perturbations. Also, higher contrast and sharpness of the in situ hybridization signal would improve the quality of these panels.

We have now increased the size of the whole-mount in situ hybridization pictures. Regarding the contrast and sharpness, we have been processing already the pictures to the best that we could in terms of contrast and sharpness. We believe that the illustrator file will give a much better resolution than what is presented as PNG in the manuscript.

Minor Issues

- Supplementary Fig. 4 contains very important results showing that misexpression of both *Lbx1* and *Btrc* is required to develop a SHFM phenotype. Therefore, this Figure should be included as a main Figure in the text. Given that authors only have 6 main Figures in the text, Suppl Fig. 4 should be moved into the main text.

We thank the reviewer for this comment and we have now moved this figure among the main ones (now Fig. 6).

- Line 262-264: The Authors should further discuss the relevance of the myogenic genes being abnormally activated in the epithelial cells of *Dup* and *Inv1* mutant mice.

The myogenic genes activation was mainly observed in the *Dup* mutant and in AER cells (Figure 5C-D). We do observe a similar trend using GO analysis from our newly generated *Inv1* single-cell data but to a lesser extent than in the *Dup* (Supplementary Fig. 6C). This can be explained by the fact that many genes are misregulated in the *Inv1* mutant (Supplementary Table 1) due to the loss of *Fgf8* expression, hiding any other effects due to *Lbx1* ectopic expression. We discussed this in lines 414-418. We do not infer this is the cause of the phenotype as the *Dup* mutant doesn't have an obvious SHFM3-like phenotype. However, we suggest that misexpression of a gene in a different tissue is able to activate a different pathway, which could contribute to the resulting phenotype.

REVIEWERS' COMMENTS

Reviewer #1 (Remarks to the Author):

The authors have answered all the questions of this reviewer.

Reviewer #3 (Remarks to the Author):

The Authors considerably reorganized the overall sequence of the findings reported in the original manuscript according to the suggestions they were given. This required a substantial amount of work including swapping of figures and adjustments to the text, but it improved the rationale and the logical sequence of the described findings, which definitely led to a clearer and tighter narrative.

In addition, the Authors went the extra mile to include additional experiments so as to address all of the critiques that were raised and all the issues that needed clarifications. For example, they improved the visualization of their 4C-seq data in human fibroblast to make ectopic contacts clearer. The revised representation and visualization of their data are now more convincing and support the presence of ectopic interactions, although subtle, involving LBX1 and BTRC in patients with the Duplication.

The Authors also provide new single cell data also for the Inv1 mutant. Interestingly, these new data show that the ectopic expression of Btrc is stronger in the Inv1 mutant than in the Dup, which adds new and robust quantitative information to the observations originally made by in situ hybridization.

The study is now stronger and tighter, the figures are elegant and masterfully crafted, and, more importantly, the results reported contribute new insights into the molecular mechanisms that underpin SHFM3, a severe limb malformation.